# C-H-activated C*sp*² -C*sp*³ diastereoselective gridization enables ultraviolet-emitting stereo-molecular nanohydrocarbons with mulitple H⋯H interactions

Ying Wei[1,6], Chunxiao Zhong [1,6], Yue Sun[1], Shuwei Ma[1], Mingjian Ni[2], Xiangping Wu[1], Yongxia Yan[1], Lei Yang[1], Ilya A. Khodov [3], Jiaoyang Ge[1], Yang Li[1], Dongqing Lin[1], Yongxia Wang[1], Qiujing Bao[1], He Zhang[1], Shasha Wang[1], Juan Song[1], Jinyi Lin [2] ✉, Linghai Xie [1,4,5] ✉ & Wei Huang[1,4,5] ✉

Gridization is an emerging molecular integration technology that enables the creation of multifunctional organic semiconductors through precise linkages. While Friedel-Crafts gridization of fluorenols is potent, direct linkage among fluorene molecules poses a challenge. Herein, we report an achiral Pd-PPh₃-cataylzed diastereoselective (>99:1 d.r.) gridization based on the C-H-activation of fluorene to give dimeric and trimeric windmill-type nanogrids (DWGs and TWGs). These non-conjugated stereo-nanogrids showcase intramolecular multiple H⋯H interactions with a low field shift to 8.51 ppm and circularly polarized luminescence with high luminescent dissymmetry factors (|$g_{PL}$| = 0.012). Significantly, the nondoped organic light-emitting diodes (OLEDs) utilizing *cis-trans*-TWG1 emitter present an ultraviolet electroluminescent peak at ~386 nm (CIE: 0.17, 0.04) with a maximum external quantum efficiency of 4.17%, marking the highest record among nondoped ultraviolet OLEDs based on hydrocarbon compounds and the pioneering ultraviolet OLEDs based on macrocycles. These nanohydrocarbon offer potential nanoscafflolds for ultraviolet light-emitting optoelectronic applications.

Macrocyclization plays a crucial role in the innovation of molecular nanohydrocarbon (MNHCs)[1-3], host-guest systems[4,5], mechanically interlocked molecules[6,7], covalent/metal-organic frameworks[8-10], as well as nanoporous materials[11]. Especially, MNHCs with strains, such as carbon nanohoops[1], nanobelts[2] and nanocages[3], have been recently created with distinctive electronic structures and diverse applications in nanoscience and materials science[1,12] thanks to the development of effective closure methodologies. However, due to the strain energy, extended π-conjugation and intense intermolecular π-π interactions, these MNHCs face challenges in achieving ultraviolet (UV) luminescence, limiting their application in the field of UV optoelectronic devices[13-16]. To the best of our knowledge, the utilization of MNHCs in

[1]Centre for Molecular Systems and Organic Devices (CMSOD), State Key Laboratory of Organic Electronics and Information Displays & Institute of Advanced Materials (IAM), Nanjing University of Posts & Telecommunications, 9 Wenyuan Road, Nanjing 210023, China. [2]Key Laboratory of Flexible Electronics (KLOFE) & Institute of Advanced Materials (IAM), Nanjing Tech University (NanjingTech), 30 South Puzhu Road, Nanjing 211816, China. [3]G.A. Krestov Institute of Solution Chemistry, Russian Academy of Sciences, Akademicheskaya str. 1, Ivanovo 153045, Russian Federation. [4]Frontiers Science Center for Flexible Electronics (FSCFE), MIIT Key Laboratory of Flexible Electronics (KLoFE), Northwestern Polytechnical University, Xi'an 710072, China. [5]School of Flexible Electronics (SoFE) and Henan Institute of Flexible Electronics (HIFE), Henan University, 379 Mingli Road, Zhengzhou 450046, China. [6]These authors contributed equally: Ying Wei, Chunxiao Zhong. ✉e-mail: iamjylin@njtech.edu.cn; iamlhxie@njupt.edu.cn; vc@nwpu.edu.cn

efficient UV organic light-emitting diode (OLED) (emission peak <400 nm) has not yet been reported, with only a limited number of acyclic aromatic hydrocarbons having been demonstrated for UV-OLEDs[15,16]. Additionally, the competition between linear polymerization and macrocyclization has also made the synthesis of MNHCs more challenging. These provide numerous opportunities for developing efficient macrocyclization strategies to synthesize MNHCs with specific properties, especial for extending their application in organic optoelectronics.

Differing from the the conventional macrocyclization methods employed in the construction of covalent organic frameworks[8,9,17], two-dimensional materials[18], and classical macrocycles[19], gridization represent an unique macrocyclization approach to building organic semiconductors. This method offers advantages such as flexibility, simplicity, and scalability, providing benefits not found in traditional macrocyclization and fusion cyclization. Until now, the achievement of gridization involves the sophisticated utilization of the 2/9-position of fluorene, thereby creating nanogrids that enable the potential linkage of 7-position sites. Significantly, through the application of the Friedel-Crafts gridization (FCG) strategy, various $Csp^3$-linked nanogrids[20–22], multi-grids[23], and even organic nanopolymer material systems[24–26] have been successfully synthesized. However, the introduction of molecular fragments containing thiophene and carbazole, leads to the fluorescence emission of the nanogrid primarily into the visible light range. Conversely, the direct connection of all carbon-based strained sub-nanometer nanogrids through the 2,9-positions functionalization of fluorene is anticipated to realize the robust UV emission. Furthermore, nanogrids, characterized by the presence of chiral carbons, offer promising avenues for chiroptical investigations[27–29] including circular dichroism (CD), circularly polarized luminescence (CPL), and luminescent dissymmetry factors ($g_{PL}$). However, despite the successful stereoselective control of diazafluorene-based drawing hand nanogrids achieved by superelectrophile-assisted FCG[30], accomplishing stereoselective synthesis of all carbon-based strained nanogrids via FCG remains a challenge. Therefore, exploring effectively stereoselective $Csp^2$-$Csp^3$ coupling methods is a highly sought-after goal for the UV-luminescent MNHCs by cooperative multiple $Csp^2$-$Csp^3$ gridization.

Metal-catalyzed C-C coupling reactions are a powerful tool not only for the $Csp^2$-$Csp^3$ gridization[31], but also for achieving stereoselective control. This is particularly achievable by flexibly tuning the metal species and ligands, specifically employing chiral transition metal[32]/ligand[33]-controlled C-H activation reactions. However, compared to expensive chiral transition metals/ligand, achieving chirality control using achiral transition metals/ligand catalyzed C-H activation is more challenging. Herein, we employed achiral transition metal-ligand-catalyzed C-H gridization reactions for the diastereoselective synthesis of all carbon-based dimeric and trimeric windmill-type nanogrids (DWGs and TWGs) from bromo-substituted H1B1-type synthons (Fig. 1). Each DWGs and TWGs have two diastereomers, with maximum molecular strain energies (MSE) of ~21.4 kcal/mol. This strategy enabled the synthesis of *cis-trans*-TWGs with high diastereoselectivity, ranging from 4.3:1 to >99:1d.r. Our results demonstrate that the achiral Pd-PPh₃-controlled C-H gridization provides an effective strategy for synthesizing strained DWGs and TWGs, with the metal Pd and PPh₃ playing crucial roles in the diastereoselectivity of the strained nanogids. We also analyzed, calculated, visualized, and revealed their molecular strains origins through space repulsion, attraction interactions and diastereomeric effects through computational and experimental methods. The investigations of photophysical properties suggested that these nanogrids exhibit UV emission, and DWGs show gridization-induced and crystal-induced luminescence enhancement features. At the same time, TWGs only display crystal-induced luminescence enhancement. Additionally, *cis-trans*-TWG1 exhibits a high |$g_{PL}$| (0.012) compared to other chiral organic small-

molecules. Notably, the non-doped UV-OLED based on the traditional fluorescence emission mechanism of *cis-trans*-TWG1 not only demonstrates a high external quantum efficiency (EQE) of 4.17% but also has an emission peak wavelength in the UV region at 386 nm with a Commission Internationale de L'Eclairage (CIE) of (0.17, 0.04), representing the pioneering UV-OLED based on macrocycles.

## Results

### Investigation of gridization rules

We commenced our investigation based on our previous work[34] by using H1B1-type substrate 2-bromo-9-phenylfluorene (**1a**) and Pd(dba)₂ catalyst in toluene at 100 ℃ (Fig. 2a, Supplementary Table 1). Through optimization (Supplementary Table 1, entries 1–9), we found that a concentration of 40 mM resulted in a 32% yield of TWG1 with 82:18 *cis-trans*-TWG1/*cis-cis*-TWG1 selectivity (Fig. 2b and Supplementary Figs. 1, 2). This suggests that the palladium complexes can eliminate the effect of strains during the gridization process. Notably, we also observed linear by-products, such as the debrominated product (**2a**, 17%), dimerized product (**3a**, 10%), trimerized product (**4a**, 7%), and linear polymer (LP1, 33%, Fig. 2b), but without trimeric nanogrid. Furthermore, other ligands, such as dppf, PCy₃, or P(*t*-Bu)₃ (Supplementary Table 1, entries 10–12) resulted in ungridized products, indicating that PPh₃ was the most effective ligand and played an essential role in the formation of TWGs. Substituting *t*-BuOK with KN(SiMe₃)₂ did not significantly affect the yield and selectivity of TWG1(Supplementary Table 1, entry 13), while no TWG1 was observed by using KOH (Supplementary Table 1, entry 14).

Given the chain-length effect on macrocyclization[35,36], we examined the chain- and arm-length effect of substrates on the yield and diastereoselectivity of $Csp^2$-$Csp^3$ gridization under optimized conditions (Fig. 2c). As the chain-length of substituents at 4-position of phenyl (**1a, 1c, 1d** and **1e**) increased, the yield of TGWs decreased from 31% to ~0%. In contrast, the *cis-trans*-TWGs/*cis-cis*-TWGs selectivity increased from 82:18 to >99:1. These results indicate that the long alkyl chain on an aryl group is unfavorable for the formation of strained nanogrids. Despite an increase in chain-length caused by introducing an oxygen atom, the yield of TGW2 increased to 25% with a *cis-trans*-TWG2/*cis-cis*-TWG2 selectivity of >99:1 compared to **1c**. These results suggest that the oxygen atom of a methoxyl group plays an essential role in the gridization process. However, when an octyloxyl group was used, the reaction gave the trimerized product (**4 f**, 13%) rather than TWGs, indicating that the steric hindrance effect of the octyloxyl group is not conducive to the formation of TWGs. Furthermore, when the aryl groups were transformed from phenyl to thienyl, the yield (37%) and *cis-trans*-TWG7/*cis-cis*-TWG7 selectivity (83:17) of TWG7 still followed the chain-length effect. As for arm-length effect (Fig. 2d), we introduced *m*-phenyl into the 2-position of fluorene (BrFPh). As a result, substrate **1 h** without a substituent group yielded strained DWG1 and TWG8 in 13% (31:69 *meso*-DWG1/*rac*-DWG1 selectivity) and 24% (32:68 *cis-trans*-TWG8/*cis-cis*-TWG8 selectivity) yields, respectively, and even larger size gridization products. These indicate that extending the arm-length can reduce the minimum nanogrids from trimer to dimer and promote the formation of larger size nanogrids, including TWGs. Moreover, when an octyloxy group (**1i**) was introduced, strained nanogrid DWG2 was not observed, but a higher yield of TWG9 was obtained at 27%, consistent with the chain- and arm-length effects. Overall, the chain- and arm-length significantly affect the yield and selectivity of strained nanogrids.

### Stereoselective origin

To investigate the origin of diastereoselectivity, we monitored the reaction. We observed that the trimerized intermediate (TPhFBr) gradually disappeared while TWGs formed (Supplementary Fig. 3) during the gridization process. This suggests the TPhFBr undergoes oxidation addition, C-H activation, and reduction elimination to form TWGs

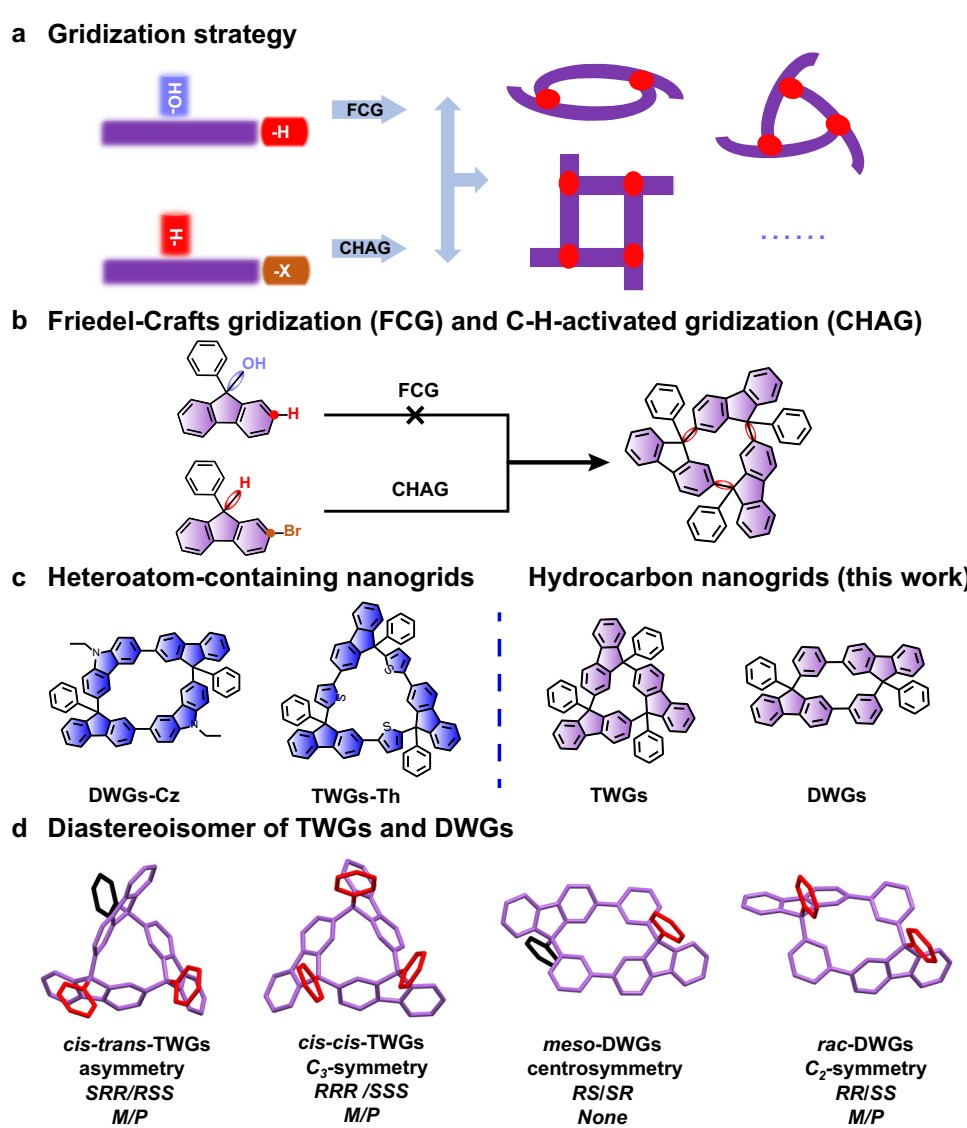

**Fig. 1 | Gridization strategy and nanogrid structures. a** A cartoon of the gridization strategy. (Ellipses indicate other multicomponent nanogrid homologues). **b** Gridization strategy of fluorene-based nanogrid. **c** The nanogrid sructures of the previous work and this work. **d** Diastereoisomers of TWGs and DWGs. (The phenyls marked as red and black exhibit phenyl at the same side or at the other side, respectively).

(Fig. 3a). Importantly, during this reaction process, there may exist four transition states, namely TWG-TS1, TWG-TS2, TWG-TS3, and TWG-TS4, which constitute a cyclic molecular entity composed of three phenyl-substituted fluorenes and palladium coordinated with two PPh$_3$ as shown in the proposed buildup scheme (Supplementary Fig. 4). We conducted density functional theory (DFT) calculations on two crucial transition states (TWG-TS1, TWG-TS2) at the B3LYP/6-31 G(d) level. The results indicate that the single-point energy of TWG-TS2 was 23.1 kcal mol$^{-1}$ lower than that of TWG-TS1 due to the stronger π-π interaction between the phenyl at 9-position of fluorene and one of the phenyl rings of PPh$_3$ (Fig. 3b). In this context, irrespective of the single-point energies of TWG-TS3 and TWG-TS4, the reaction would preferentially produce *cis-trans*-TWG1 rather than *cis-cis*-TWG1. Moreover, the π-π interaction on TWG-TS2 is significantly enhanced by the C-H···O interaction between the oxygen atom of the methoxybenzyl and the phenyl of PPh$_3$. When the substrate 2-bromo-9-octylfluorene was examined, no TWG was observed (Fig. 3c), indicating that aromatic rings at the 9-position of fluorene affected *cis-trans*-TWGs selectivity but also served as an essential prerequisite for generating TWGs. However, the transition states of larger than trimeric nanogrids may be difficult to form due to the significant steric

hindrance within the molecules. Thus, the formation of tetrameric and larger nanogrids cannot be observed. Increasing the arm-length may help reduce steric hindrance in the transition states and promote the formation of larger nanogrids.

## Structural characterization of DWGs and TWGs

The structures of diastereomers of DWG1 and TWG1 were demonstrated by high resolution mass spectrum (HRMS), (Supplementary Figs. 5, 6), NMR spectra (Fig. 3d, Supplementary Figs. 9–13). The disappearance of the peak at 5.03 ppm in the $^1$H NMR spectra and the signal at 54.4 ppm in the $^{13}$C NMR spectra confirmed substrate conversion into DWGs and TWGs via gridization. Interestingly, a distinctive phenomenon was observed in the $^1$H NMR spectra of *cis-trans*-TWG1 and *cis-cis*-TWG1. In the $^1$H NMR spectrum of *cis-trans*-TWG1, the chemical shift of H(8') shifted to high-field (singlet peak at 6.00 ppm) due to the strong shielding of adjacent fluorene. Conversely, the chemical shift of proton H(8) in the $^1$H NMR spectrum of *cis-cis*-TWG1 dramatically shifts to 8.19 ppm due to the double deshielding of two adjacent fluorenes and two phenyl groups linked at the 9-position (supported by the $^1$H-$^1$H NOESY cross-peak H(8)/H(a), H(e) in Fig. 4a), as well as van der Waals interactions (H···H delta (Δ)-attraction

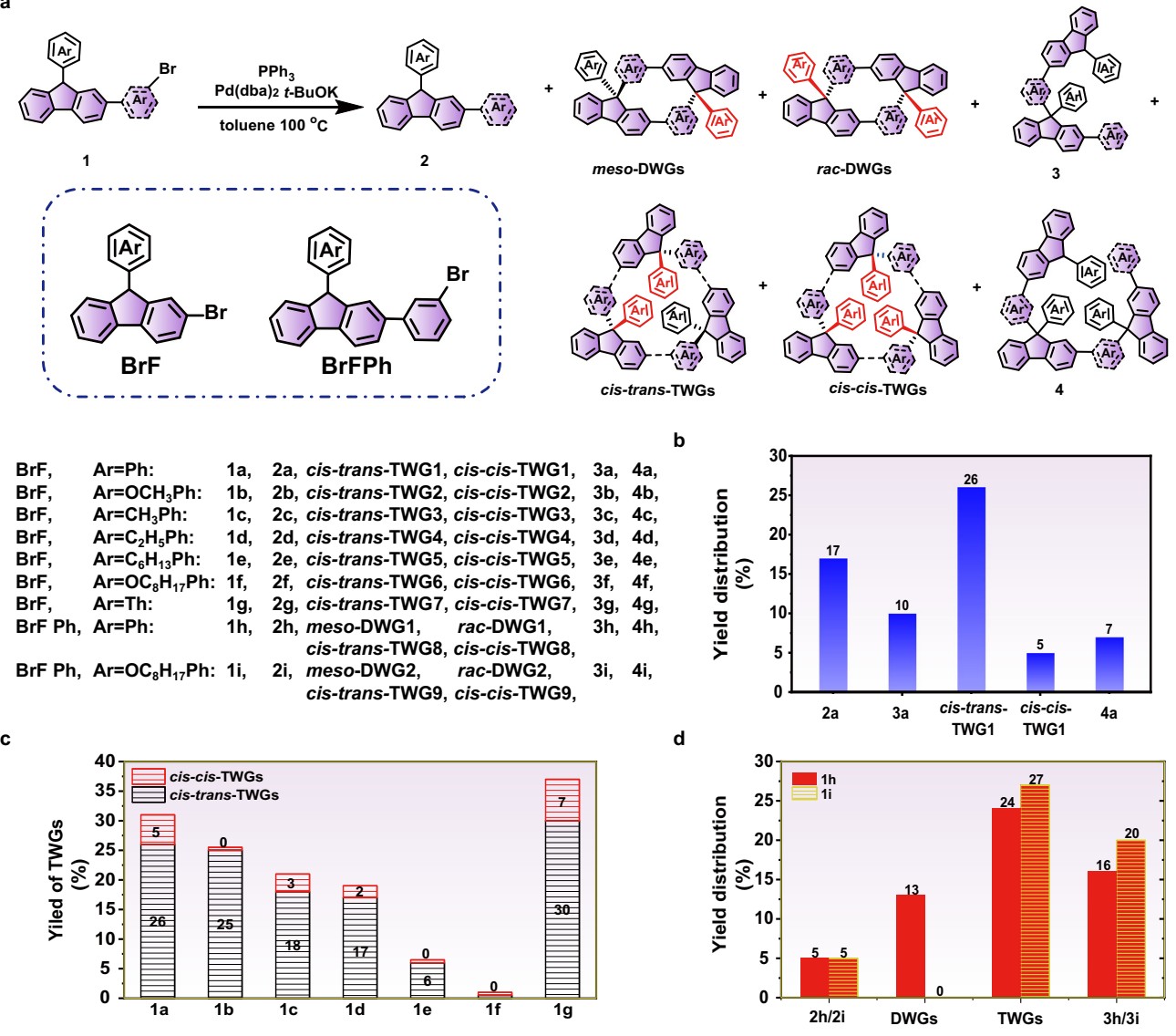

**Fig. 2 | Characteristics on the C-H-activated gridization. a** Corresponding formula for C-H-activated gridization of **1a-1i**. **b** C-H-activated gridization yield distribution of **1a**. **c** Effect of substituents on aryl groups on TWGs yield and TWGs selectivity. **d** C-H-activated gridization yield distribution of **1 h** and **1i**.

interaction, see below) that are similar to those observed in *cis-trans*-TWG1 with the cross-peak H(8)/H(8″) (Fig. 4a). Substituting the phenyl at the 9-position with 2-thiophenyl caused the chemical shift of proton H(8) to shift even further to 8.51 ppm due to the S···H interaction between H(8) and thiophene (Supplementary Fig. 14). This characteristic signal was not observed in other fluorene-based nanogrids and macrocycles. The chemical shift of H(8) in these compounds typically fell within the range of 6.73 to 8.11 ppm (Supplementary Fig. 15)[13,20–22,36–39]. It should be noted that this unique signal could only be achieved through the combined effect of deshielding and van der Waals interactions, as only one of these effects alone would not produce the same result. For instance, the same signal was not observed in the ¹H NMR spectra of *meso*-DWGs and *rac*-DWGs, where only one of these factors was present. Specifically, the structure of *meso*-DWGs had only an H···H ▱-attraction interaction, while the structure of *rac*-DWGs had only a deshielding effect (Supplementary Fig. 16).

**Noncolvant interaction and strain analysis of DWGs and TWGs**

To further elucidate the precise structures, weak interactions and molecular strains of DWGs and TWGs, we used single-crystal X-ray diffraction and non-covalent index visualization techniques.

Fortunately, we obtained single crystals via slow evaporation in dichloromethane/isopropanol (*cis-trans*-TWG1) or dichloromethane/*n*-hexane (*cis-cis*-TWG1, *meso*-DWG1 and *rac*-DWG1) (Fig. 4b). Co-crystals of *cis-trans*-TWG1/*cis-cis*-TWG1 was also obtained in tetrahydrofuran/water (Supplementary Fig. 17). Specifically, the diastereoisomers had similar geometric sizes, with *cis-trans*-TWG1 and *cis-cis*-TWG1 measuring 1.08 nm, 1.13 nm, and 1.13 nm, and 1.11 nm, 1.11 nm, and 1.13 nm, respectively. Similarly, *meso*-DWG1 and *rac*-DWG1 exhibited geometric sizes of 0.77 nm and 1.09 nm and 0.70 nm and 1.11 nm, respectively. Moreover, considering the effect of weak interaction on the luminescence properties, we also measured the distances between H···H, π···π, and C-H···π to elucidate the variations in weak intramolecular interactions among the diastereoisomers of DWGs and TWGs. Firstly, within the H···H interaction range of 0.74–3.04 Å[40], H···H interactions mainly exist between H(4) and H(5) of a fluorene unit (*cis-trans*-TWG1: 2.69 Å, 2.70 Å and 2.71 Å; *cis-cis*-TWG1: 2.72 Å, 2.75 Å and 2.70 Å) and the adjacent hydrogen of 8-position among fluorenes (*cis-trans*-TWG1: 2.18 Å; *cis-cis*-TWG1: 2.19 Å, 2.19 Å and 2.31 Å) in single crystal (Supplementary Table 8). In *cis-cis*-TWG1, a symmetric H···H delta (Δ)-attraction interaction was observed with three H···H interactions among the adjacent hydrogen at the 8-position of fluorenes

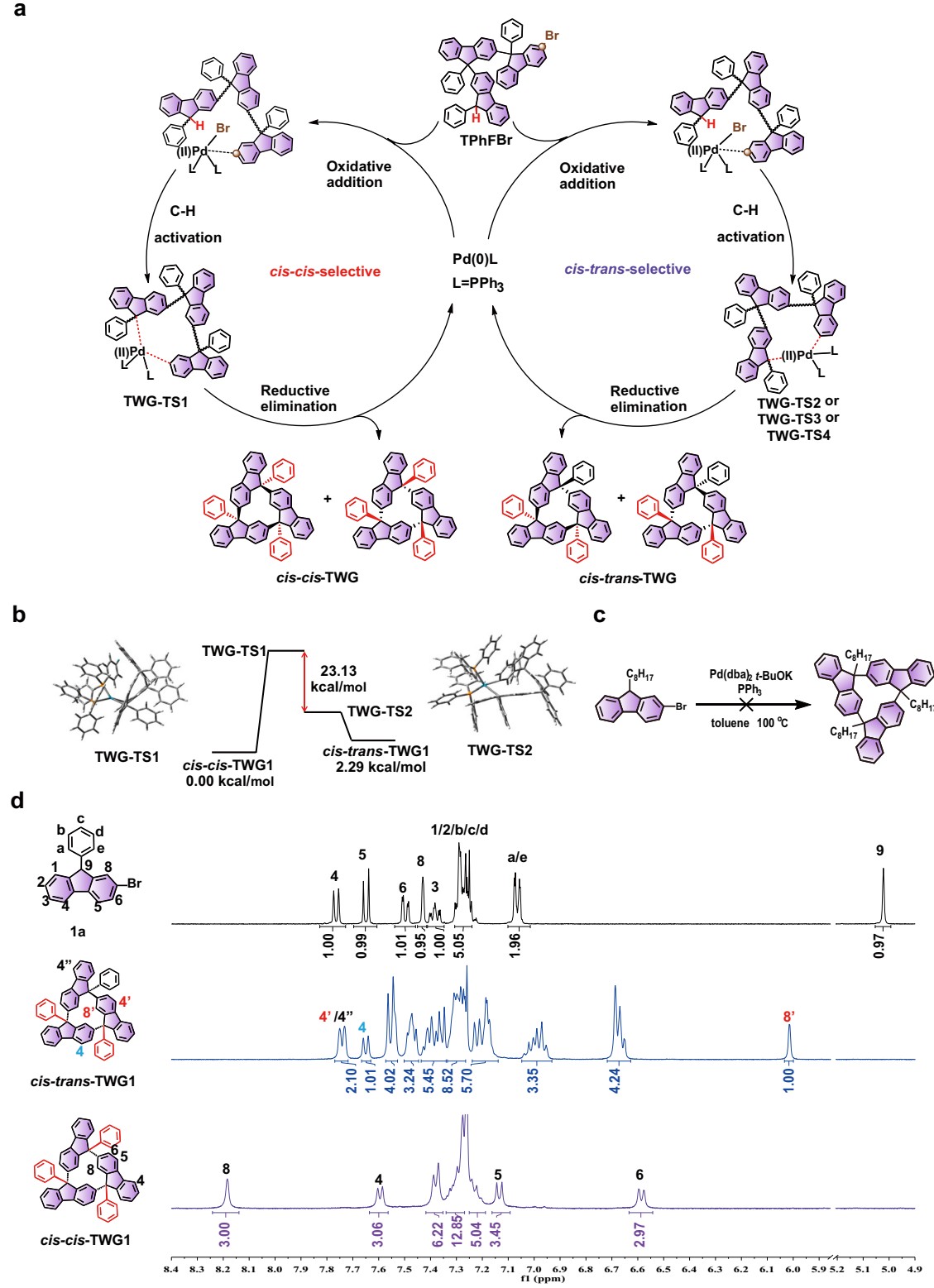

**Fig. 3 | Mechanistic investigations and the ¹H NMR characterization. a** Proposed mechanism diagram of TWGs. **b** Intermediate diagram of theoretical calculation. **c** Corresponding formula for C-H gridization of 2-bromo-9-octylfluorene. **d** The ¹H NMR spectra of **1a**, *cis-trans*-TWG1 and *cis-cis*-TWG1.

(Supplementary Fig. 18). Similarly, in the single-crystal structures of *meso*-DWG1 and *rac*-DWG1 (Supplementary Table 9), H···H interactions mainly exist between 1-position of fluorene and 15-position of phenyl on the backbone (*rac*-DWG1: 2.24 Å, 2.25 Å; *meso*-DWG1: 2.14 Å, 2.14 Å) and the adjacent hydrogen of 8-position among fluorenes and 11-position phenyl on the backbone (*rac*-DWG1: 2.07 Å, 2.26 Å; *meso*-

DWG1: 2.24 Å, 2.24 Å, 2.20 Å, 2.20 Å). In *meso*-DWG1, a symmetric H···H ▱-attraction interaction was observed with four H···H interactions among the adjacent hydrogen at the 8-position of fluorenes and 11-position phenyl on the backbone (Supplementary Fig. 16). Besides, a π···π Δ-repulsion interaction was observed with π···π stacking interactions on the vertexes between adjacent fluorenes (Supplementary

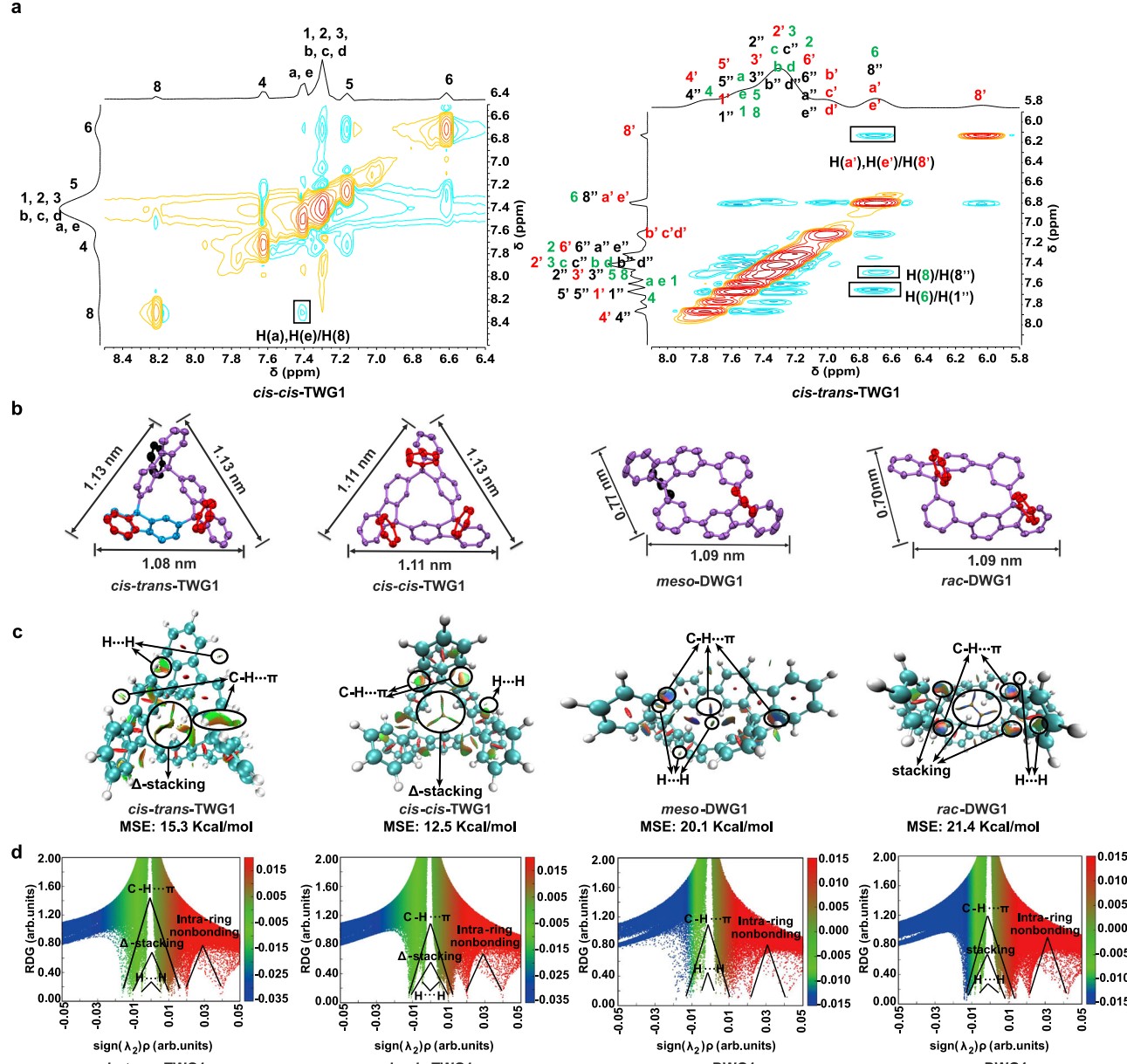

**Fig. 4 | Structural characterization and non-colvant interaction analysis of TWGs and DWGs. a** 2D $^1$H-$^1$H NOESY spectra of *cis-cis*-TWG1 and *cis-trans*-TWG1. (Arabic numerals and letters of the same color represent hydrogen protons located within the same 9-phenylfluorene unit.) **b** The single-crystal crystallography of *cis-trans*-TWG, *cis-cis*-TWG, *meso*-DWG1 and *rac*-DWG1. (The red and black lines represent phenyl at the same side or the other side, respectively). **c** The non-covalent interactions (NCI) isosurfaces for *cis-trans*-TWG1, *cis-cis*-TWG1, *meso*-DWG1 and *rac*-DWG1. **d** Plots of the reduced density gradient (RDG) of DWG1 and TWG1.

Fig. 18) for both *cis-trans*-TWG1 (3.18 Å, 3.04 Å and 3.08 Å) and *cis-cis*-TWG1 (3.06 Å, 3.06 Å and 3.05 Å, Supplementary Table 10) in a single crystal. In addition to these H⋯H interactions, diastereomers also exhibited other weak intermolecular interactions, including C-H⋯π and π⋯π interactions (Supplementary Tables 10–12). Finally, these weak interactions were demonstrated by 3D reduced density gradient (RDG) isosurfaces with blue-green-red (BGR) color scales representing the sign($\lambda_2$)ρ values and the scatter plot of sign($\lambda_2$)ρ versus RDG (Fig. 4c, d, and Supplementary Figs. 19, 20).

Compared to other nanogrids, nanorbelt and [18]CPP[30,41–45], the DWGs and TWGs have higher MSE values, but lower than [12]CPP[45] (Supplementary Fig. 21). Here, the strains of DWGs and TWGs were analyzed in detail, including angle strain, bending strain and torsion strain, using a single crystal. Firstly, the angle of C$sp^3$ is an

important parameter to describe the angle strain (Supplementary Tables 13, 14). The results show that the angle strain of *meso*-DWG1 and *cis-cis*-TWG1 (average angle strain: 3.54° and 3.77°, respectively) is slightly larger than that of *rac*-DWG1 and *cis-trans*-TWG1 (1.76° and 1.44°, respectively). The angle strain of DWGs and TWGs is generally the same, but much smaller than that of cyclopropane (49.5°)[46]. Secondly, the bending strain was analyzed by measuring the out-of-plane distortion, which is the distance of a backbone unit (Supplementary Tables 15, 16). The result of single crystal XRD for *meso*-DWG1, *rac*-DWG1 and *cis-cis*-TWG1 show that all the backbone units are stretched. However, two fluorene units are stretched, and one is bent in the backbone units of *cis-cis*-TWG1. Furthermore, the bending strain of *meso*-DWG1 and *rac*-DWG1 (average out-of-plan distortion: 0.16 Å and 0.19 Å, respectively) is more significant than that of *cis-cis*-TWG1 and

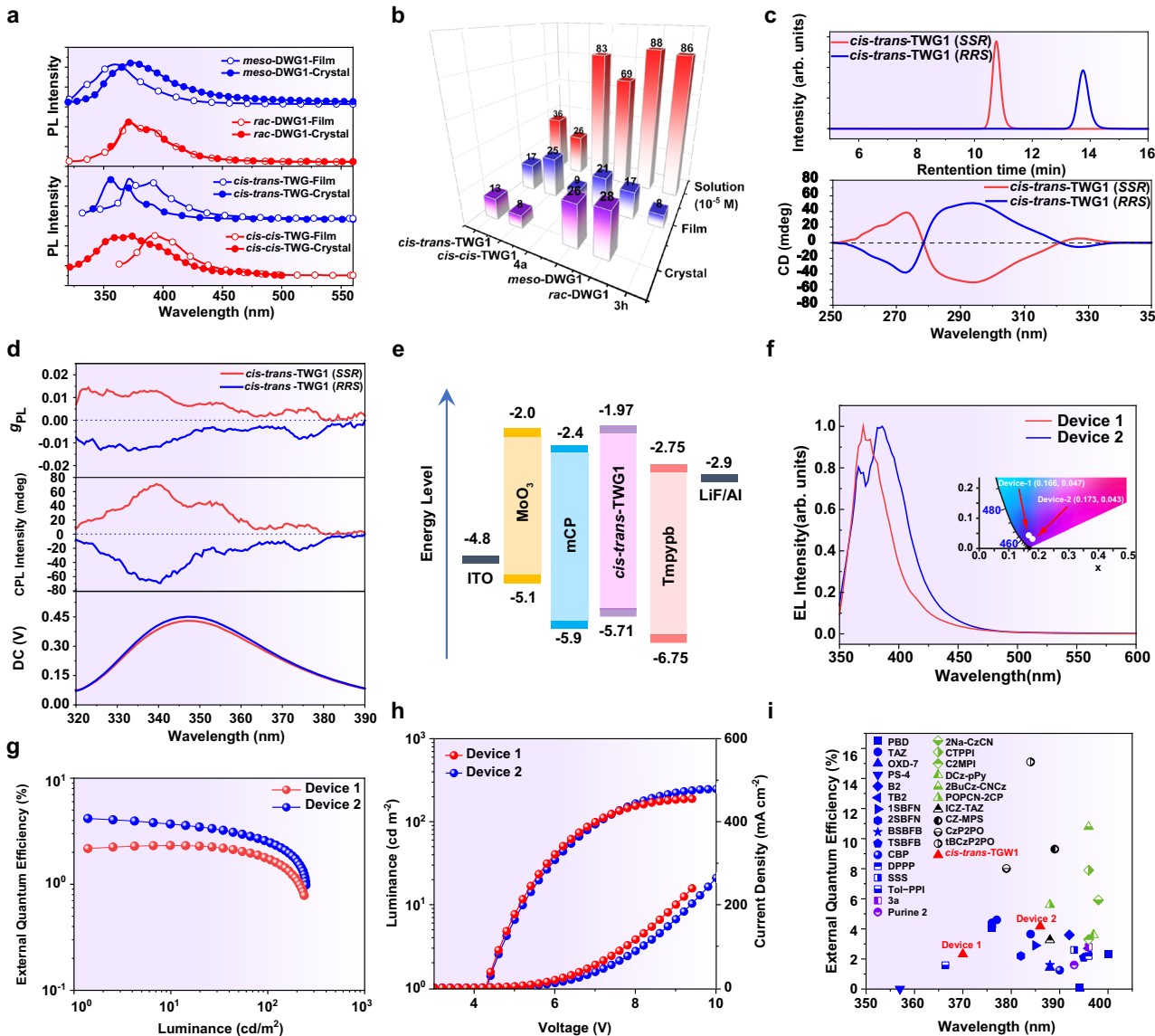

**Fig. 5 | Ultraviolet emission properties of hydrocarbon nanogrids. a** PL spectra of TWG1 and DWG1 in film and crystal. **b** PLQY of **3 h, 4a**, TWG1 and DWG1 in the THF solution ($C = 10^{-5}$ M), film and crystal states. **c** Chiral HPLC analysis of *cis-trans-*TWG1 eluted by MeOH/DCM (4:1) using CHIRALPAK IJ column and circular dichroism spectra of the first peak (*cis-trans*-TWG1 (*SSR*)) and the second peak (*cis-trans*-TWG1 (*RRS*)) in toluene solution ($C = 5 \times 10^{-6}$ M). **d** CPL spectra of *cis-trans-*TWG1 (*SSR*) and *cis-trans*-TWG1 (*RRS*) in toluene solution ($C = 5 \times 10^{-6}$ M) under an excitation wavelength of 290 nm. **e** Configuration of the *cis-trans*-TWG1-based UV OLED. **f** EL spectrum of Device 1 and Device 2 based on *cis-trans*-TWG1. Inset shows the corresponding coordinates on CIE1931 chromaticity plane. **g** External quantum efficiency-current density curve and **h** the current density-voltage-luminance (*J-V-L*) characteristics of Device 1 and Device 2 based on *cis-trans*-TWG1 UV OLED. **i** External quantum efficiency summary of the representative UV OLEDs (EL Peak ≤400 nm). Blue represents non-doped traditional fluorescence OLED; Purple represents doped traditional fluorescence OLED; Green represents hybridized local and charge-transfer (HLCT) based OLEDs; Black represents thermally activated delayed fluorescence (TADF) OLED.

*cis-trans-*TWG1 (0.06 Å and 0.01 Å, respectively). The bending strain of DWGs is nearly the same as [4]C-diBu-F (0.19 Å) and [5] C-diBu-F (0.1 Å)[36]. Thirdly, the torsion strain is measured by the external torsion angle between two adjacent units of the nanogrid (Supplementary Tables 17, 18). Here, the torsion strain of *cis-trans*-TWG1 (average external torsion angle: 61.82°) is more significant that of *cis-cis*-TWG1, *meso*-DWG1 and *rac*-DWG1 (33.13°, 37.58° and 35.37°, respectively.). Moreover, another torsion strain exists between fluorene and phenyl on the backbone unit for DWG1 (average external torsion angle: *meso*-DWG1, 35.12° and *rac*-DWG1, 35.15°).

## Ultraviolet emission properties of hydrocarbon nanogrids

After confirming the molecular structure, we investigated optical properties of DWGs and TWGs in solution and film states (Fig. 5a, b,

Supplementary Table 19). In diluted solution, the absorption spectra of DWG1 and TWG1 indicated that their optical bandgap (3.70, 3.59, 3.63 and 3.74 eV for *meso*-DWG, *rac*-DWG1, *cis-cis*-TWG1 and *cis-trans*-TWG1, respectively) were slightly lower than that of ungridized **3 h** (3.71 eV) and **4a** (3.80 eV) (Supplementary Figs. 24, 28), respectively, and the photoluminescence spectra of these nanogrids exhibit redshift compared with the corresponding linear molecules (Supplementary Figs. 25, 29). These trends can be attributed to the strained feature[47] and spiroconjugation of DWG1 as well as TWG1, and also indicate that the gridized backbone enhances the behaviors of excitonic delocalization[25]. However, the emission peaks of DWG1 and TWG1 (361, 372, 393 and 372 nm for *meso*-DWG1, *rac*-DWG1, *cis-cis*-TWG1 and *cis-trans*-TWG1, respectively) are still in the UV-region in film state with the CIE coordinate of (0.17, 0.05), (0.16, 0.01), (0.16, 0.02) and (0.16, 0.03)

(Supplementary Table 19), respectively, owing to their non-conjugated linkage, which is different from conjugated nanohoops[1] that exhibit redshifting to the visible region. Notably, both DWG1 and TWG1 showed gridization-induced luminescence enhancement feature, with higher photoluminescence quantum yield (PLQY) in the nanogrids (21% for *meso*-DWG1, 17% for *rac*-DWG1, 25% for *cis-trans*-TWG1, 17% for *cis-cis*-TWG1) than in the corresponding linear molecules (8% for **3 h**, 9% for **4a**) (Fig. 5b). This may be because the robust and steric multi-dimensinal topological structure to suppress intermolecular interactions of strained nanogrids and further inhibit the non-radiative relaxation[48]. After that, the aggregate behaviors make a correlation with the UV emission behaviors. Specifically, the crystal of *meso*-DWG1 displayed red-shifted luminescence (from 361 to 374 nm) with the CIE coordinates of (0.16, 0.09) (Supplementary Table 19) when compared to the amorphous films, while *rac*-DWG1 showed no change. However, the crystals of both *cis-cis*-TWG1 and *cis-trans*-TWG1 exhibited blue-shifted emission (from 393 to 373 nm for *cis-cis*-TWG1, and from 372 to 356 nm for *cis-trans*-TWG1) compared to the amorphous films (Fig. 5a). The differences in the aggregated emission between DWG1 and TWG1 may be attributed to their different molecular packing modes in the crystalline state (Supplementary Figs. 32, 33)[49]. Additionally, compared to amorphous films, the PLQY of single crystals increased from 21 to 26% for *meso*-DWG1 and from 17 to 28% for *rac*-DWG1. In comparison, it decreased from 25 to 8% for *cis-cis*-TWG1 and from 17 to 13% for *cis-trans*-TWG1 (Fig. 5b). These observations suggested that DWG1 exhibited crystal-induced luminescence enhancement feature[50–53], while TWG1 did not, probably due to its tighter arrangement in the crystalline (the density of single crystals: 1.335 gcm$^{-3}$, 1.296 gcm$^{-3}$, 1.238 gcm$^{-3}$ and 1.081 gcm$^{-3}$ for *meso*-DWG, *rac*-DWG1, *cis-cis*-TWG1 and *cis-trans*-TWG1, respectively) and strong intermolecular interactions, effectively suppressing the non-radiative relaxation process. Interestingly, these results contradict the phenomenon of red-shifted and quenched emission observed in most luminescence due to exciton coupling, orbital overlap, and π–π aggregation as pressure increases[54].

Considering the fact that the nanogrid itself contains chiral carbon at 9-postion of fluorene units, there is research value in studying its chiroptical properties after separating enantiomers. Fortunately, we successfully separated *cis-trans*-TWG1 using chiral high-performance liquid chromatography (HPLC) with MeOH/DCM = 80/20 (V/V) as the mobile phase, resulting in two distinct peaks (P1 and P2). Comparative analysis based on measured CD spectra and simulated data (Supplementary Fig. 46) indicates that P1 corresponds to *cis-trans*-TWG1 (*SSR*), while P2 corresponds to *cis-trans*-TWG1 (*RRS*) (Fig. 5c and Supplementary Fig. 47). Considering the fluorescence properties of *cis-trans*-TWG1, we further characterized the CPL properties of the enantiomers *cis-trans*-TWG1 (*SSR*) and *cis-trans*-TWG1 (*RRS*) in toluene solution (Fig. 5d). The results reveal strong CPL signal at 340 nm for both *cis-trans*-TWG1 (*SSR*) and *cis-trans*-TWG1 (*RRS*). Notably, the |$g_{PL}$| value of *cis-trans*-TWG1 was calculated to be 0.012 (Fig. 5d)[55], surpassing the majority of organic chiral small molecules (typically at the 10$^{-4}$–10$^{-3}$ level)[27–29].

In general, our steric hydrocarbon nanogrids present pure UV emission behavior in solid states. Especially, their intrinsically multidimensional steric interactions may be beneficial for the suppression of intermolecular π-interaction to obtain robust emission behavior, which are useful to manufacture the efficient non-doped UV-OLEDs. As discussion above, TWG1 coated films had a relatively high UV emission efficiency than those of the DWG1 ones, so we set *cis-trans*-TWG for optoelectronic application. Therefore, to verify their high potential application for use in electroluminescent (EL) device, non-dopded OLEDs are fabricated with a configuration of ITO/ MoO₃/ mCP/Nanogrids EML/ETL/LiF/Al (120 nm) (Fig. 5e). Here, molybdenum oxide (MoO₃) serves as the hole injection layer, 9,9′-(1,3-phenylene)bis-*9H*-carbazole (mCP) and 1,3,5-tris(3-pyridyl-3-phenyl) benzene (Tmpypb)

are employed as the hole and electron transport layers (HTLs and ETLs), respectively, and lithium fluoride (LiF) acts as a buffer layer to enhance the device's electron injection capability. *cis-trans*-TWG1 is utilized as the emitting layer (EML). Interestingly, device (1) based on *cis-trans*-TWG1 achieved the excellent color purity with a pure UV maximum EL emission peaked at 370 nm with narrow full width at half maximum (FWHM) as small as 34 nm, and corresponding to the ideal CIE coordinate of (0.17, 0.05), locate at the near-UV edge of the chromaticplane (Fig. 5f). These results also effectively confirmed the promising applicaiton of our steric hydrocarbon nanogrids in UV nondoped OLEDs.

Meanwhile, we further optimized *cis-trans*-TWG1-based OLEDs structure trying to improve the device efficiency, observed that the ETL can slightly change the EL peak wavelength and FWHM, further slightly change the color purity, due to the optical microcavity effect (Fig. 5f)[56]. The device efficiency is signficantly influence by the electron affinity of the ETL and emitter materials, which may modulate the charge injection and recombination in the EML. According to the energy level as displayed in the Fig. 5e, the selection of mCP and Tmpypb as HTL and ETL nearly equal to *cis-trans*-TWG1 is beneficial for enhancing the barrier-free hole/electron injection and transportation. Besides, we tuned the film thickness of mCP, such as 45 nm (Device 1), and 55 nm (Device 2), as displayed in Fig. 5f, the EQE value of device are increasing from 2.32%, to 4.17% (Fig. 5g). Large recombination zone may be observed for a thicker HTL layer in the OLEDs to induce this enhancement of device efficiency (EQE). And correspoding the turn-on voltages (V$_{on}$) are estimated about 4.4 V, and 4.6 V with a maximum brightness of 189 cd/m², and 248 cd/m² (Fig. 5h), respectively. Therefore, as we discussion above, the *cis-trans*-TWG1 endowed its UV nondoped OLEDs with the state-of-the-art EQE up to 4.17% (Fig. 5g). The record EQE may be attributed to the synergistic effect of its improved out-coupling ratio and suppressed intermolecular interaction, exciton utilization efficiency, and suitable energy levels. We also believed that the efficiency of OLEDs based on steric hydrocarbon nanogrids can be further improved via systematically optizing device structure and doping strategy. More intereisngly, deivce efficiency (EQE) of typical UV-OLEDs (EL peak <400 nm) are summaried in Fig. 5i and Supplementary Table 22. Surprisingly, TAZ type molecules is few UV fluorescent emitter for the efficient non-doped OLED (EQE > 3.6%, CIE: 0.17, 0.04)[57], which can further improve to EQE > 4.14% need to introduce special hole injection layers for the realization of the balance between hole and electron injection/transpoortation[58–60]. Therefore, the EQE of our *cis-trans*-TWG1-basd OLEDs is the record value achieved by traditional non-doped fluorescence UV-OLEDs based on the hydrocarbons compounds, which is the pioneering UV-OLEDs based on macrocycles (Fig. 5i).

## Discussion

In conclusion, we present an achiral Pd-PPh₃-controlled diastereoselective Csp²-Csp³ gridization, allowing the efficient one-pot synthesis of strained TWGs and DWGs. This approach provides high diastereoselectivity in synthesizing TWGs, attributed to the stronger π-π interaction between the benzene ring at 9-position of fluorene and one of the benzene rings of PPh₃ in the transition states. Additionally, we also observed unique intramolecular steric H···H *Δ*(▱)-attraction, π···π *Δ*-repulsion in TWGs and DWGs, and a shift of one of the fluorene protons to low-field in ¹H NMR spectrum, not observed in other fluorene-based compounds. Moreover, these nanogrids exhibit diverse photophysical behaviors, including the CPL property (|$g_{PL}$| = 0.012). More interestingly, efficient UV-OLEDs based on *cis-trans*-TWG1 with a narrowband emission at 386 nm are also fabricated with a highest EQE of 4.17%, representing the pioneering UV-OLEDs based on macrocycles. Ongoing studies on these nanogrids are focused on their potential applications in UV-OLEDs.

## Methods

### General information

All reactions' progress was monitored using thin-layer chromatography to ensure they had reached completion. Gas chromatographic analyses were conducted on a Varian GC 2000 gas chromatography instrument equipped with an FID detector, using biphenyl as the internal standard. Some products underwent purification through column chromatography over silica gel (200–300 mesh) obtained from Nanjing Wanqing Chemical Instruments Company, while others were purified via recrystallization. Prior to use, all solvents were dried and distilled following standard methods. Cycle preparation liquid chromatography (CPLC) was performed on LaboACE LC-5060, while high-performance liquid chromatography (HPLC) was conducted using Agilent technologies. Unless specifically stated, materials were acquired from commercial suppliers and used without additional purification. Circularly Polarized Luminescence data were collected using a JASCO CPL 300. Magnetic circular dichroism spectrum data were obtained using JASO Corp, J-810.

### Materials

The synthetic procedures and details of the Grignard reaction, dehydroxylation reaction, alkylation reaction and Pd-PPh$_3$-controlled gridization are described in the Supplementary Methods. The synthetic procedures of substrates are detailed in the Supplementary Methods as well.

### Structural characteristics via NMR technologies

The bruker 400 MHz Fourier Transform NMR spectrometer was used to obtain $^1$H and $^{13}$C NMR spectra at a frequency of 400 MHz and 100 MHz, respectively, at 20 °C. The coupling constants are expressed in Hz as a J value. The 90° pulses and a relaxation delay of 1 s were employed to record $^1$H NMR spectra with a spectral width of 12.02 ppm and 128 scans. $^{13}$C NMR spectra were recorded using 45° pulse sequence with power-gated decoupling for suppression of the scalar couplings with protons, a spectral width of 236 ppm, 1024 scans, and a relaxation delay of 1 s. The values of chemical shifts are represented by δ in units of parts per million (ppm) relative to an internal standard [$^1$H NMR: tetramethylsilane (TMS) = 0.00 ppm] or relative residual peaks ($^1$H NMR: 7.26 ppm for CDCl$_3$; $^{13}$C NMR: 77.0 ppm triplet for CDCl$_3$). Briefly, the multiplicities of each signal peak are represented as s (singlet), d (doublet), t (triplet), q (quartet), dd (doublet of doublets), dt (doublet of triplets), and m (multiplet). Two-dimensional nuclear Overhauser effect spectroscopy (2D ge-NOESY) [https://doi.org/10.1016/j.molliq.2022.120525; https://doi.org/10.1016/j.molliq.2022.120481] experiments were conducted with pulsed filtered gradient techniques [https://doi.org/10.1006/jmra.1996.0222], utilizing a phase-sensitive mode with 2048 points in the F2 direction and 512 points in the F1 direction, a mixing time value of 0.30 s, and 24 scans with a relaxation delay of 2 s.

### Measurements of molar mass with mass spectrometry

Molecular mass of substrates was determined using Matrix-assisted laser desorption/ionization time-of-flight mass spectrometry (MALDI-TOF-MS). High-resolution mass spectra (HRMS) were recorded on Bruker solanX 70 FT-MS.

### Single-crystal crystallography

Data of *cis-trans*-TWG1, *cis-trans*-TWG1/*cis-cis*-TWG1, *meso*-DWG1, and *rac*-DWG1 single-crystal crystallography was recorded at around 100 K using a Bruker 2000 CCD area detector and graphite-monochromated Mo Kα radiation (λ = 0.71073 Å).

### DFT calculations

All the quantum chemical calculations were performed using Gaussian 09 Program package. Hybrid-density functional theory (DFT) calculations with the B3LYP and 6-31 G(d) basis set were used to explore the geometrical structure of all of the stationary points (the isomers and transition states) and electronic properties(noncolvant interaction and molecular strain energy) of Ph, *meso*-DWG1, *rac*-DWG1, DPh, DF, TF, SWG-F, Th, DTh, TWG-Th, **2**, **3**, *cis-trans*-TWG1, *cis-cis*-TWG1, TWG-TS1 and TWG-TS2. Non-covalent interaction analysis (NCI) was performed using Multiwfn [https://doi.org/10.1002/jcc.22885] and VMD software [https://doi.org/10.1016/0263-7855(96)00018-5] to observe the weak intramolecular interactions. The fluorescence emission spectrum of TWGs was calculated by time-dependent density functional theory (TD-DFT) under the SMD continuum solvent model for tetrahydrofuran. The geometric structure optimization of *cis-trans*-TWG1(*SSR*) and *cis-trans*-TWG1(*RRS*) was performed using the Cam-B3LYP/6-31 g method in Gaussian 09 suite of programs, while the circular dichroism spectra of *cis-trans*-TWG1(*SSR*) and *cis-trans*-TWG1(*RRS*) were computed via TD-DFT at Cam-B3LYP/TZVP level.

### Photophysical behaviors

Photoluminescence quantum yields of *cis-trans*-TWG1, *cis-cis*-TWG1, **4a**, *meso*-DWG1, *rac*-DWG1, **3 h** were recorded on Hamamatsu C9920-02G. UV absorption spectra of *cis-trans*-TWG1, *cis-trans*-TWG1, *cis-cis*-TWG1, **4a**, *meso*-DWG1, *rac*-DWG1, **3 h** were recorded on LAMBDA 35 spectrophotometer. Fluorescence emission spectra of *cis-trans*-TWG1 were recorded on RF-6000 Plus spectrophotometer.

### Fabrication of OLEDs

The ITO substrates were successively cleaned in ultrasonic bath of acetone, isopropanol, detergent and deionized water, respectively. After totally drying in a 70 °C oven, the ITO substrates were treated by O$_2$ plasma to improve the hole injection ability. The vacuum-deposited OLEDs were fabricated under a pressure of less than $5 \times 10^{-4}$ Pa in the Fangsheng OMV-FS380 vacuum deposition system. The organic layers, LiF and Al were deposited at rates of 1–2, 0.1, and 5 Å s$^{-1}$, respectively.

### Measurement of OLEDs

The active area of each device was 3 mm × 3 mm. The *L-V-J* characteristics and EL spectra of UV OLEDs were measured by a Konica Minolta CS-200 Color and Luminance Meter and an Ocean Optics USB 2000+ spectrometer, along with a Keithley 2400 Source Meter. For OLEDs, the L-V-J characteristics, EL spectra were obtained via a PhotoResearch PR670 spectroradiometer, with a Keithley 2400 Source Meter. The EQEs were estimated assuming that the devices are Lambertian emitters. All characterizations were done at room temperature under ambient conditions without any encapsulation.

## Data availability

The authors declare that all data supporting the current findings of this study are available in the main manuscript or in the Supplementary information. The single crystal datum (including a cif. file and a structural figure) of *cis-trans*-TWG1, *cis-cis*-TWG1, *cis-trans*-TWG1/*cis-cis*-TWG1, *meso*-DWG1, *rac*-DWG1 were uploaded to Cambridge Structural Database, obtaining corresponding CCDC number 2052303, 2124396, 2052304, 2248637, and 2248641 respectively. The data related to the figures and other findings of this study are available from the corresponding author upon request. https://doi.org/10.6084/m9.figshare.25296160.

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

## Acknowledgements

This work was supported by the Natural Science Foundation of China (62288102, 22071112, 22275098 and 22075136), the National Key R&D Program of China (No. 2020YFA0709900), the Project of State Key Laboratory of Organic Elec-tronics and Information Displays, Nanjing University of Posts and Telecommunications, (GDX2022010005 and GZR2023010001), the open research fund from the State Key Laboratory of Luminescent Materials and Devices (South China University of Technology); We also greatly acknowledged the Professor Zhiming Wang and Miss Jingli Lou from South China University of Technology for their professional experiment and suggestions in fabrication of OLEDs.

## Author contributions

Ying Wei has provided significant suggestions on the synthesis properties analysis of DWGs and TWGs, and written this manuscript and Supplementary Materials. Chunxiao Zhong has provide date analysis of DWGs and TWGs, and written this manuscript. Yue Sun and Shuwei Ma have written this Supplementary Materials, synthesized the DWGs and TWGs and tested photophysical characterizations. Xiangping Wu has performed the Pd-PPh$_3$-controlled gridization, grown and confirmed single crystals and analyzed strain energies of TWG1. Yongxia Yan has synthesized the DWGs. Ilya A. Khodov has provided the help in NMR spectra. Yang Li and Dongqing Lin have provided the help in UV-*vis* absorption and PL spectra. Qiujing Bao has helped to address the figures. Ge jiaoyang has helped verify strain energies data. He Zhang and Shasha Wang have provided the help in single crystal analysis. Juan Song has provided the help in C-H activation reaction. Mingjian Ni and Jinyi Lin fabricated the OLED and performance evaluation. Yongxia Wang and Lei Yang have provided crucial help in theoretical calculation. Linghai Xie, Jinyi Lin and Wei Huang has initiated this project, provided crucial idea and offers enough funds for this work. Ying Wei (†) and (†) Chunxiao Zhong were contributed equally.

## Competing interests

The authors declare no competing interests.
