## [Peer Review File · Nature Communications]

REVIEWER COMMENTS

Reviewer #1 (Remarks to the Author):

This paper by Xie and Huang et al. describes a macrocycle synthesis based on C-H activation for Csp²-Csp³ diastereoselective bond formation. This paper is the latest work in the grid-like macrocycle synthesis project using fluorene as a building block, which is being vigorously pursued by the authors. The macrocycle synthesis in this paper is novel in that it uses a Pd-catalyzed C-H activation method instead of their conventional macrocycle synthesis method that relies on the Friedel-Crafts reaction. It is also interesting to note that high diastereoselectivity is obtained. The structures of the cyclic dimer and trimer obtained in the synthesis are unique. On the other hand, the diastereoselectivity is not well discussed. In particular, the reaction pathway is discussed based on DFT calculations, but a rational explanation of how the pathway was found is not described, despite the many conformations and steps involved. As for the macrocycle properties, there is a strong claim to UV luminescence properties. However, this reviewer could not find the distinctive advantage of this macrocycle in UV luminescence properties. The luminescence quantum yield is not necessarily high. It should be discussed, for example, what the level is for appropriate comparative materials that could be expected to have laser applications, etc (a little is in Supplementary Fig. 22). As described above, the synthesis itself is interesting, and the structure of the obtained molecule is unique. Still, the lack of attractive physical properties makes this study inappropriate for publication in this journal.

Reviewer #2 (Remarks to the Author):

In this manuscript, the authors reported a new gridization protocol, which used an achiral Pd-PPh₃ catalyst and reached a high diastereoselectivity. Such a protocol is based on C-H-activation at the fluorene the 9-position, which produces dimeric and trimeric windmill-type nanogrids. Properties of the products were investigated with various means. My main concern is about the broad interest which is required to be published in Nature Communications. The authors mentioned that their results may have applications for ultraviolet organic light-emitting diodes and lasing applications. But it is not clear if there will be strong impact to these areas. About the novelty, the authors should clarify if this study is the first achiral catalyzed diastereoselective gridization. Another comment is more details about the methods should be provided, such as the way to calculate the transition states.

Reviewer #3 (Remarks to the Author):

In this manuscript, Wei et al reported a stereoselective Csp²-Csp³ coupling gridization based on the C-H activation of fluorene and synthesized dimeric and trimeric

windmill-type nanogrids. The resultant nanogrids show ultraviolet emission behaviors and crystal-induced luminescence enhancement features. Combining the experimental results and calculations, the authors have clearly revealed the gridization rules, stereoselective origin, and molecular strain of resultant nanogrids.

The manuscript may be accepted given the authors address the points detailed below.

1. The products rac-DWG1, cis-trans-TWG1 and cis-cis-TWG1 have the clear chiral features and the separation seems promising in supplementary fig. 2. It is essential to report their chiroptical properties, including circular dichroism, circularly polarized luminescence properties and individual dissymmetric factors. Their unique chiroptical properties might attract widespread readership in macrocycles fields.
2. Although the products are characterized by NMR with assignments by 2D NMR and crystal structures, the purity of the products is important. In SI, page S29, figure S9, there are obvious impurity showing the peaks around 10-50 ppm. The authors emphasize that the nanogrids present first reported UV emission among macrocycles. It is especially important to have pure sample for optical properties measurements and avoid errors in reporting the properties of this family of nanogrids.
3. The manuscript and SI might need to be polished. It is not easy to follow their expression. In addition, there are some mistakes in the manuscript that should be corrected. For example, in the section of investigation of gridization rules, "we found that a concentration of 40 mM resulted in a 32% yield of TWGs1 with 82:18". In Supplementary Table 1, the result should be 35% yield with Dr ratio of 80:20. In the same section, "Substituting t-BuOK with KN(SiMe₃)₂ did not significantly affect the yield and selectivity of TWG1(Supplementary Table 1, entry 13), while no TWG1 was observed by using KOH (Supplementary Table 1, entry 14)." According to Supplementary Table 1, it should be entry 14 and 15, respectively.
4. Please replace the supplementary fig. 5-8 with high-resolution mass spectra.

Respond to Reviewers

We sincerely thank the reviewers for carefully reviewing our work and the constructive and insightful comments, which have helped us to improve the paper further. Our detailed responses to the reviewer's comments are provided in the following. Changes in the revised manuscript as a response to the reviewers' comments are highlighted in *red color* and clarifications regarding the reviewer's comments are provided in *blue color*.

Respond to Reviewers:

Reviewer #1:

This paper by Xie and Huang et al. describes a macrocycle synthesis based on C-H activation for Csp²-Csp³ diastereoselective bond formation. This paper is the latest work in the grid-like macrocycle synthesis project using fluorene as a building block, which is being vigorously pursued by the authors. The macrocycle synthesis in this paper is novel in that it uses a Pd-catalyzed C-H activation method instead of their conventional macrocycle synthesis method that relies on the Friedel-Crafts reaction. It is also interesting to note that high diastereoselectivity is obtained. The structures of the cyclic dimer and trimer obtained in the synthesis are unique. On the other hand, the diastereoselectivity is not well discussed. In particular, the reaction pathway is discussed based on DFT calculations, but a rational explanation of how the pathway was found is not described, despite the many conformations and steps involved.

Answer: Thank you for your careful comments and suggestions. To investigate the origin of diastereoselectivity, we have proposed a buildup scheme, as illustrated in Fig. R1. First, substrate 1a undergoes a C-H activation reaction to generate dimeric intermediates B1 and B2. Subsequently, these dimeric intermediates undergo C-H activation reactions again via paths I-VI to produce trimeric intermediates TPhFBr₁, TPhFBr₂, TPhFBr₃, and TPhFBr₄. Finally, these trimeric intermediates go through four potential cyclic transition states (TWG-TS1, TWG-TS2, TWG-TS3, and TWG-TS4) to yield the gridization products *cis-cis*-TWG and *cis-trans*-TWG. Based on density functional theory (DFT) calculations at the B3LYP/6-31G(d) level, we calculated the single-point energies of two important transition states, TWG-TS1 and TWG-TS2. The results show that the strong π - π interaction between the phenyl group at the 9-position of fluorene and one benzene ring of PPh₃ significantly lowers the single-point energy of TWG-TS2 by 23.1 kcal mol⁻¹ compared to TWG-TS1. Specifically, taking dimer B1 as an example, it has three possible reaction pathways—namely, paths I, II, and III. For path II, regardless of the single-point energy of

TWG-TS4, the ultimate product is always *cis-trans*-TWG. Regarding paths I and III, if the single-point energy of TWG-TS3 is lower than that of TWG-TS1, the reaction is more likely to yield *cis-trans*-TWG. However, if the single-point energy of TWG-TS3 is greater than or equal to TWG-TS1, the significantly lower energy of TWG-TS2 compared to TWG-TS1 gives path II a reactive advantage over paths I and III, leading to a preference for *cis-trans*-TWG over *cis-cis*-TWG. Similarly, dimer B2 exhibits the same preferred reaction pathways as B1. Therefore, regardless of the single-point energies of TWG-TS3 and TWG-TS4, the lower energy of TWG-TS2 compared to TWG-TS1 directs the reaction towards pathways favoring the generation of *cis-trans*-TWG, resulting in high diastereoselectivity.

Fig. R1. The proposed buildup scheme of TWG and the structural formula of transition states.

Moreover, the π - π interaction on TWG-TS2 is significantly enhanced by the C-H...O interaction between the oxygen atom of the methoxybenzyl and the phenyl group of PPh₃.

When the substrate 2-bromo-9-octylfluorene was examined, no TWG formation was observed, indicating that aromatic rings at the 9-position of fluorene not only influence *cis-trans*-TWGs selectivity but also serve as an essential prerequisite for TWG generation. Finally, based on these findings, we propose a plausible reaction mechanism (Fig. R2).

Fig. R2. Proposed mechanism diagram of TWG

As for the macrocycle properties, there is a strong claim to UV luminescence properties. However, this reviewer could not find the distinctive advantage of this macrocycle in UV luminescence properties. The luminescence quantum yield is not necessarily high. It should be discussed, for example, what the level is for appropriate comparative materials that could be expected to have laser applications, etc (a little is in Supplementary Fig. 22).

Answer: Thank you for your careful suggestions on UV luminescence properties of the nanogrids. Firstly, molecular nanohydrocarbon (MNHCs) like carbon nanorings, nanobelts, and nanocages face challenges in achieving ultraviolet fluorescence emission due to factors such as molecular skeleton tension, strong π -conjugation, and intense intermolecular π - π

interactions. We successfully achieved ultraviolet fluorescence emission by employing a molecular design strategy that limits the π -conjugation length through direct connection at the 2/9 positions of fluorene with conjugation interruption and introduces steric hindrance groups at the 9-position of fluorene. Notably, this strategy maintains ultraviolet fluorescence emission, especially in thin film and crystalline states. Secondly, in the thin film state, the lattice structure, compared to the linear structure, exhibits increased molecular rigidity and enhanced intramolecular weak interactions due to gridization. This contributes to the suppression of non-radiative relaxation and an improvement in luminescence efficiency (*Nat. Commun.* **13**, 2850 (2022)). Our test results also indicate that the photoluminescence quantum yields (PLQY) of grids are higher than the corresponding linear structures. On the other hand, to generate broader interest among readers, in accordance with the reviewer's suggestions, we initially supplemented the study on the chiroptical properties of *cis-trans*-TWG1. We separated *cis-trans*-TWG1 into a pair of enantiomers, *cis-trans*-TWG1 (*RRS*) and *cis-trans*-TWG1 (*SSR*), using MeOH/DCM=80/20(V/V) as the mobile phase through chiral high-performance liquid chromatography (HPLC) (Table R1). Subsequently, we characterized their toluene dilute solutions (5×10^{-6} M) for chiroptical properties, including circular dichroism (CD) and circularly polarized luminescence (CPL) (Fig. R3). Based on the measured CD spectra and simulated CD spectra, it was observed that peak 1 corresponds to *cis-trans*-TWG1 (*SSR*), while peak 2 corresponds to *cis-trans*-TWG1 (*RRS*). Notably, both *cis-trans*-TWG1 (*RRS*) and *cis-trans*-TWG1 (*SSR*) exhibit exceptional circularly polarized luminescent properties, with a luminescence anisotropy factor as high as 0.01, surpassing that of the majority of chiral small molecules (*ChemPhotoChem* 2018, **2**, 386–402; *ChemPhotoChem* 2022, **6**, e202100228; *J. Mater. Chem. C*, 2023, **11**, 2053–2062; *CCS Chem.* 2023, **5**, 2760-2789).

Table R1 | Separation method of *cis-trans*-TWG1 in HPLC.

Column	: CHIRALPAK IJ
Column size	: 5.0 cm I.D. × 25 cm L, 10 μ m
Mobile phase	: MeOH/DCM=80/20(V/V)
Flow rate	: 30 ml/min
Wave length	: UV 225 nm
Temperature	: 38 °C

Moreover, we are agreed with you that we need to systematically explore the potential application in ultraviolet light-emitting optoelectronic devices. According to your comment and suggestion, we also modified improper sentence and explanation about their potential application in light-emitting devices in this revised manuscript.

In general, the luminescence quantum yield of conjugated molecules is the key but not only one factor to modulate the efficiency of light-emitting optoelectronic devices. For example, according to many previous works (*Chem. Rev.* 2016, **116**, 12823–12864. *J. Am. Chem.*

Soc. 2015, **137**, 15105-15111. *Cryst. Growth Des.* 2009, **9**, 4945-4950. *Adv. Mater. Interfaces* 2020, **7**, 1902057), organic emitters with a relatively low emission efficiency (PLQY < 30%) also present an excellent lasing behavior. Beyond emission efficiency (PLQY), the optical loss, intrinsic refractivity and morphological quality of organic gain materials, together with the quality of the resonant cavity, are the key parameters to obtain low threshold organic lasers. More interestingly, to isolate the novel macrocycles in the blend matrix is another effective strategy to enhance emission efficiency and photo-stability for the fabrication of ultraviolet organic laser, which are confirmed by large number previous works (*Chem. Rev.* 2016, **116**, 12823–12864. *Chem. Soc. Rev.*, 2020, **49**, 5885-5944. *Nat Commun* 2015, **6**, 8458. *ACS Macro Lett.* 2016, **5**, 967-971. *J. Mater. Chem. C* 2013, **1**, 1182-1191). In the last decades, fluorene-based organic materials present a robust lasing behavior at ultraviolet and deep regions from 350~480 nm (*Chem. Rev.* 2016, **116**, 12823–12864. *Chem. Soc. Rev.*, 2020, **49**, 5885-5944). Therefore, we believed that our novel MNHCs are expected to have laser applications. Similarly, the efficiency of organic light-emitting diodes (OLEDs) is closely associated with the emission efficiency, out-coupling ratio and exciton utilization efficiency. Based on the discussion, we reasonably predicated that our novel MNHCs may be act as promise application in light-emitting optoelectronic devices.

Meanwhile, as your comments, we firstly tried to investigate the lasing behavior of our TWG and DWG in solid states. Up to date, the lasing behavior of organic molecules can be realized via optical excitation. Therefore, in the last several months, we made our attempts to probe the possible lasing behavior of our molecules. Due to the large bandgap of our novel molecules, the absorption spectra of our novel are reasonably ranged from 250~320 nm with a maximum peak at 300 nm. Unfortunately, we cannot find the lasing source with a peak at <325 nm. Therefore, we cannot further investigate the lasing behavior of our novel materials. More interestingly, as an alternative light-emitting optoelectronic device to organic lasers, OLEDs is a promising device for the commercial optoelectronic applications. Therefore, we also fabricated OLED based on our novel *cis-trans*-TWG1. As we expected, **efficient and narrowband UV OLEDs are successfully manufactured with a highest EQE of 4.17%**. This is record efficiency value of ultraviolet OLED based on the pure hydrocarbon emitters. Device also present a narrowband emission with a FWHM of 34 nm and CIE of (0.17, 0.04), locate at the near-UV edge of the chromaticplane. We also added the discussion in this revised manuscript. Therefore, we are easily confirmed that our stereo-molecular nanohydrocarbons showed a greatly potential application in light-emitting optoelectronic devices.

In the last several decades, there is rare report to prepare organic molecules for the fabrication of the efficient UV OLEDs with peak wavelength below 400 nm, due to their high energy of ultraviolet photon and wide bandgap. Recently, multiple resonances planar organic molecules for TADF devices are attracted more attentions for efficient and narrowband doped ultraviolet OLEDs, associated with low vibration relaxation and molecular perturbation, stable molecular configuration between ground and excited states

(*Nature Photonics* **13** (2019): 678-682. *Angewandte Chemie International Edition*, 2022, **61**(48): e202209425. *Aggregate*, 2022, **3**(2): e144. *Angewandte Chemie International Edition*, 2023, **62**(46): e202312666. *Advanced Optical Materials*, 2022, **10**(22): 2201714.). Besides, TAZ type molecules is few UV fluorescent emitter for the efficient non-doped OLED (EQE > 3.6%, CIE: 0.17, 0.04) (*Organic Electronics* 2020, **82**, 105718), which can further improve to EQE > 4.14% need to introduce novel hole injection layers for the realization of the balance between hole and electron injection/transportation (*Journal of Materials Chemistry C* 2019, **7**, 926-936. *Applied Physical Letters* 2017, **110**, 043301. *Organic Electronics* 2017, **46**, 7-13). In fact, it is reasonably predicated that the fragile C-N, C-P and C-S bonds in the organic conjugated molecules may result into their and OLEDs instability (*Angew. Chem.* 2022, **134**, e202207204. *J. Appl. Phys.* 2007, **101**, 024512. *J. Phys. Chem. C* 2012, **116**, 19451–19457. *J. Phys. Chem. C* 2014, **118**, 7569–7578.), especial for the ultraviolet and blue OLEDs. Then, hydrocarbons aromatic emitters also had a intrinsically stability to resist high band exciton for stable ultraviolet and blue OLEDs, benefits of the removal of heteroatoms to avoid the weak and chemical bond in the chromophore, similar to the commercialized anthracene (*Nat Commun* 2023, **14**, 3927. *Angew. Chem.* 2022, **134**, e202207204). To the best of our knowledge about previously reported narrowband non-doped ultraviolet OLEDs (CIE_y ≤ 0.04), the great majority showed the EQE lower than 4%. In addition, compared to the planar and fused aromatic emitters, non-planar conjugated molecules with a multi-dimensional topical structure showed an extremely weak intermolecular aggregation and interaction to obtain single-molecular emission behavior with low ratio of non-radiative transitions (*NPG Asia Materials*, 2021, **13**(1): 53.). Therefore, it is very excited to obtain a narrowband and efficient UV non-doped OLEDs based on the novel steric hydrocarbon *cis-trans*-TWG1. Therefore, it is reasonably believed that our novel steric MNHCs have a promising application in the efficient and stable UV OLEDs. We also provided discussion in detail about the fabrication of UV OLED in this revised manuscript as follow:

“In general, our novel rigid and steric MNHCs present pure UV emission behavior in solid states. Especially, their intrinsically multi-dimensional steric interactions may be beneficial for the suppression of intermolecular π -interaction to obtain robust emission behavior, which are useful to manufacture the efficient non-doped UV organic light-emitting diodes (OLEDs). As discussion above, TWG coated films had a relatively high UV emission efficiency than those of the DWG ones, so we set *cis-trans*-TWG for optoelectronic application. Therefore, to verify their high potential application for use in electroluminescent device, non-doped OLEDs are fabricated with a configuration of ITO/ MoO₃/ mCP/Nanogrids EML/HEL/LiF/Al (120 nm) (Fig. 5e). Here, molybdenum oxide (MoO₃) serves as the hole injection layer, 9,9'-(1,3-Phenylene)bis-9H-carbazole (mCP) and 1,3,5-Tris(3-pyridyl-3-phenyl) benzene (Tmppyb) are employed as the hole and electron transport layers, respectively, and lithium fluoride (LiF) acts as a buffer layer to enhance the device's electron injection capability. *cis-trans*-TWG1 is utilized as the emitting layer (EML). Interestingly, **Device 1** based on *cis-trans*-TWG1 achieved the excellent color purity with a

pure UV maximum EL emission peaked at 370 nm with narrow full width at half maximum (FWHM) as small as 34 nm, and corresponding to the ideal CIE coordinate of (0.166, 0.047), locate at the near-UV edge of the chromaticplane (Fig. 5f). These results also effectively confirmed the promising applicaiton of our novel streic nanohydrocarbon macrocycles in UV non-doped OLEDs.

Meanwhile, we further optimized *cis-trans*-TWG1-based OLEDs structure trying to improve the device efficiency, observed that the ETL can slightly change the EL peak wavelength and FWHM, further slightly change the color purity, due to the optical microcavity effect (Fig. 5f)⁵⁶. The device efficiency is significantly influence by the electron affinity of the ETL and emitter materials, which may modulate the charge injection and recombination in the EML. According to the energy level as displayed in the Fig. 5e, the selection of mCP and Tmpypb as HTL and ETL nearly equal to *cis-trans*-TWG1 is beneficial for enhancing the barrier-free hole/electron injection and transportation. Besides, we tuned the film thickness of mCP, such as 45 nm (Device 1), and 55 nm (Device 2), as displayed in Fig. 5f, the EQE value of device are increasing from 2.32%, to 4.17% (Fig. 5g). Large recombination zone may be observed for a thicker HTL layer in the OLEDs to induce this enhancement of device efficiency (EQE). And correspondng the turn-on voltages (V_{on}) are estimated about 4.4 V, and 4.6 V with a maximum brightness of 189 cd/m², and 248 cd/m² (Fig. 5h), respectively. Therefore, as we discussion above, our novel *cis-trans*-TWG1 endowed its UV non-doped OLEDs with the state-of-the-art EQE up to 4.17% (Fig. 5g). The record EQE may be attributed to the synergistic effect of its improved out-coupling ratio and suppressed intermolecular interaction, exciton utilization efficiency, and suitable energy levels. We also believed that the efficiency of OLEDs based on our novel molecules can be further improved via systematically optizing device structure and doping strategy. More interestingly, deivce efficiency (EQE) of typical UV OLEDs (EL peak < 400 nm) are summarized in Fig. 5i. Surprisingly, TAZ type molecules is few UV fluorescent emitter for the efficient non-doped OLED (EQE > 3.6%, CIE: 0.17, 0.04)⁵⁷, which can further improve to EQE > 4.14% need to introduce novel hole injection layers for the realization of the balance between hole and electron injection/transportation.⁵⁸⁻⁶⁰ Therefore, the EQE of our *cis-trans*-TWG1-basd OLEDs is the record value achieved by traditional non-doped fluorescence UV OLEDs based on the hydrocarbons compounds, which is the first efficient UV OLEDs based on macrocycles (Fig. 5i).”

As described above, the synthesis itself is interesting, and the structure of the obtained molecule is unique. Still, the lack of attractive physical properties makes this study inappropriate for publication in this journal.

Answer: Thank you for your comments on our work.

According to your comments and suggestions, we paid more attempts to systematically explore **the photophysical properties of novel molecules, such as photophysical processing, UV-light emission, chiroptical properties together with their efficient UV electroluminescent property** in this revised manuscript. These fantastic optoelectronic

properties of our novel also effectively confirmed their greatly potential application in the organic photonic and electronics. We provided a discussion in detail here as follow:

1. Unique UV emission. This work initially demonstrates the UV luminescent characteristics of nanogrids, which have been challenging to achieve in existing MNHCs such as carbon nanohoops, nanobelts, and nanocages. The fluorescence emission of these molecules typically falls within the visible or even near-infrared range due to their inherent high skeletal tension, extended π conjugation, and strong intermolecular π - π interactions. The strategy of directly connecting the 2/9 positions of fluorene maintains molecular tension while effectively reducing molecular π conjugation. Additionally, introducing a phenyl group at the 9th position of fluorene serves as a steric hindrance group, effectively lowering intermolecular π - π interactions. The synergistic effect of these two strategies enables nanogrid-based molecular nanocarbons to successfully achieve UV fluorescence emission. Furthermore, in the thin film state, the grid structure, compared to the corresponding linear structure, enhances molecular skeletal rigidity and intra-molecular weak interactions, aiding in suppressing non-radiative relaxation and improving luminescent efficiency.

2. Fantastic chiroptical properties. Considering the fact that the nanogrid itself contains chiral carbon at 9-position of fluorene units, there is research value in studying its chiroptical properties after separating enantiomers. In terms of chiroptical properties, ***cis-trans-TWG1* exhibits outstanding luminescence anisotropy with a factor of 0.01 in a dilute solution, surpassing the luminescence anisotropy of most organic chiral small molecules (which typically range from 10^{-4} to 10^{-3}).**

3. Efficient UV OLEDs. More importantly, as anticipated, efficient and narrowband UV OLEDs have been successfully fabricated with the strongest emission peak at 370 nm and the highest EQE (external quantum efficiency) of 4.17%, comparable to UV TADF (thermally activated delayed fluorescence) devices. To the best of our knowledge, in previously reported narrowband non-doped ultraviolet OLEDs (peak ≤ 370 nm, CIE_y ≤ 0.04), the vast majority exhibited EQEs lower than 4%. **This work signifies the first application of macrocyclic structures in UV OLEDs, and the EQE of 4.17% sets a record for the efficiency of UV OLEDs based on pure hydrocarbon emitters.**

Fig. R3. Ultraviolet emission properties of hydrocarbon nanogrids. (a) PL spectra of TWG1 and DWG1 in film and crystal. (b) PLQY of 3h, 4a, TWG1 and DWG1 in the solution, film and crystal states. (c) Chiral HPLC analysis of *cis-trans*-TWG1 eluted by MeOH/DCM (4:1) using CHIRALPAK IJ column and circular dichroism spectra of the first peak (*cis-trans*-TWG1 (SSR)) and the second peak (*cis-trans*-TWG1 (RRS)). (d) CPL spectra and g_{PL} values of *cis-trans*-TWG1 (SSR) *cis-trans*-TWG1 (RRS). (e) Configuration of the *cis-trans*-TWG1-based UV OLED. (f) EL spectrum of device 1 and device 2 based on *cis-trans*-TWG1. Inset shows the corresponding coordinates on CIE1931 chromaticity plane. (g) External quantum efficiency-current density curve and (h) the current density-voltage-luminance (J - V - L) characteristics of Device 1 and Device 2 based on *cis-trans*-TWG1 UV OLED. (i) External quantum efficiency summary of the representative UV OLEDs (EL Peak \leq 400 nm). Blue represents non-doped traditional fluorescence OLED; Purple represents doped traditional fluorescence OLED; Green represents hybridized local and charge-transfer (HLCT) based OLEDs, Black represents thermally activated

delayedfluorescence (TADF) OLED.

In summary, the outstanding physical properties and performance highlighted in this work probably make it suitable for publication in *Nature Communications*.

Reviewer #2:

In this manuscript, the authors reported a new gridization protocol, which used an achiral Pd-PPh₃ catalyst and reached a high diastereoselectivity. Such a protocol is based on C-H-activation at the fluorene the 9-position, which produces dimeric and trimeric windmill-type nanogrids. Properties of the products were investigated with various means. My main concern is about the broad interest which is required to be published in Nature Communications. The authors mentioned that their results may have applications for ultraviolet organic light-emitting diodes and lasing applications. But it is not clear if there will be strong impact to these areas.

Answer: Thank you for your careful comments about this work. For broader interest, this work firstly demonstrates the synthesis of nanogrids with high enantioselectivity (>99:1 d.r) through non-chiral catalyzed C-H activation reactions. Achieving high enantioselectivity in C-H activation reactions often requires the use of expensive chiral catalysts or ligands. Therefore, this work may attract researchers involved in asymmetric synthesis and C-H activation reactions. Additionally, the newly synthesized all-carbon strained nanogrids in this work, as a novel molecular nanostructure, achieve UV fluorescence emission for the first time. This is expected to arouse the interest of researchers engaged in the synthesis and photophysical properties of carbon nanostructures, nanobelts, nanocages, and other strained aromatic macrocycles.

On the other hand, to appeal to a broader readership, we have followed the reviewer's suggestion to supplement the study of the chiroptical properties of *cis-trans*-TWG1. Through chiral high-performance liquid chromatography, we successfully separated the two enantiomers of *cis-trans*-TWG1 (*SRR*) and *cis-trans*-TWG1(*RSS*), and characterized them using circular dichroism and circularly polarized luminescence. The results show that *cis-trans*-TWG1 (*SRR*) and *cis-trans*-TWG1(*RSS*) exhibit strong CPL intensity, and their luminescence dissymmetry factors reach 0.01, surpassing the dissymmetry factors of most organic chiral small molecules. Therefore, this work may also pique the interest of researchers in chiroptical properties.

Moreover, we are agreed with you that we need to systematically explore the potential application in UV light-emitting optoelectronic devices. According to your comment and suggestion, we also modified improper sentence and explanation about their potential application in light-emitting devices in this revised manuscript.

In general, the luminescence quantum yield of conjugated molecules is the key but not only one factor to modulate the efficiency of light-emitting optoelectronic devices. For example, according to many previous works (*Chem. Rev.* 2016, **116**, 12823–12864. *J. Am. Chem. Soc.* 2015, **137**, 15105-15111. *Cryst. Growth Des.* 2009, **9**, 4945-4950. *Adv. Mater. Interfaces* 2020, **7**, 1902057), organic emitters with a relatively low emission efficiency (PLQY < 30%) also present an excellent lasing behavior. Beyond emission efficiency (PLQY), the optical loss, intrinsic refractivity and morphological quality of organic gain materials, together with the quality of the resonant cavity, are the key parameters to obtain

low threshold organic lasers. More interestingly, to isolate the novel macrocycles in the blend matrix is another effective strategy to enhance emission efficiency and photo-stability for the fabrication of ultraviolet organic laser, which are confirmed by large number previous works (*Chem. Rev.* 2016, **116**, 12823–12864. *Chem. Soc. Rev.*, 2020, **49**, 5885-5944. *Nat Commun* 2015, **6**, 8458. *ACS Macro Lett.* 2016, **5**, 967-971. *J. Mater. Chem. C* 2013, **1**, 1182-1191). In the last decades, fluorene-based organic materials present a robust lasing behavior at ultraviolet and deep regions from 350~480 nm (*Chem. Rev.* 2016, **116**, 12823–12864. *Chem. Soc. Rev.*, 2020, **49**, 5885-5944). Therefore, we believed that our novel MNHCs are expected to have laser applications. Similarly, the efficiency of organic light-emitting diodes (OLEDs) is closely associated with the emission efficiency, out-coupling ratio and exciton utilization efficiency. Based on the discussion, we reasonably predicated that our novel MNHCs may be act as promise application in light-emitting optoelectronic devices.

Meanwhile, as your comments, we firstly tried to investigate the lasing behavior of our TWG and DWG in solid states. Up to date, the lasing behavior of organic molecules can be realized via optical excitation. Therefore, in the last several months, we made our attempts to probe the possible lasing behavior of our molecules. Due to the large bandgap of our novel molecules, the absorption spectra of our novel are reasonably ranged from 250~320 nm with a maximum peak at 300 nm. Unfortunately, we cannot find the lasing source with a peak at <325 nm. Therefore, we cannot further investigate the lasing behavior of our novel materials. More interestingly, as an alternative light-emitting optoelectronic device to organic lasers, OLEDs is a promising device for the commercial optoelectronic applications. Therefore, we also fabricated OLED based on our novel *cis-trans*-TWG1. As we expected, **efficient and narrowband ultraviolet OLEDs are successfully manufactured with a highest EQE of 4.17%**. This is record efficiency value of ultraviolet OLED based on the pure hydrocarbon emitters. Device also present a narrowband emission with a FWHM of 34 nm and CIE of (0.17, 0.04), locate at the near-UV edge of the chromaticplane. We also added the discussion in this revised manuscript. Therefore, we are easily confirmed that our stereo-molecular nanohydrocarbons showed a greatly potential application in light-emitting optoelectronic devices.

In the last several decades, there is rare report to prepare organic molecules for the fabrication of the efficient ultraviolet OLEDs with peak wavelength below 400 nm, due to their high energy of ultraviolet photon and wide bandgap. Recently, multiple resonances planar organic molecules for TADF devices are attracted more attentions for efficient and narrowband doped ultraviolet OLEDs, associated with low vibration relaxation and molecular perturbation, stable molecular configuration between ground and excited states (*Nature Photonics* **13** (2019): 678-682. *Angewandte Chemie International Edition*, 2022, **61**(48): e202209425. *Aggregate*, 2022, **3**(2): e144. *Angewandte Chemie International Edition*, 2023, **62**(46): e202312666. *Advanced Optical Materials*, 2022, **10**(22): 2201714.). Besides, TAZ type molecules is few ultraviolet fluorescent emitter for the efficient non-doped OLED (EQE > 3.6%, CIE: 0.17, 0.04) (*Organic Electronics* 2020, **82**, 105718), which can

further improve to EQE > 4.14% need to introduce novel hole injection layers for the realization of the balance between hole and electron injection/transportation (*Journal of Materials Chemistry C* 2019, **7**, 926-936. *Applied Physical Letters* 2017, **110**, 043301. *Organic Electronics* 2017, **46**, 7-13). In fact, it is reasonably predicated that the fragile C-N, C-P and C-S bonds in the organic conjugated molecules may result into their and OLEDs instability (*Angew. Chem.* 2022, **134**, e202207204. *J. Appl. Phys.* 2007, **101**, 024512. *J. Phys. Chem. C* 2012, **116**, 19451–19457. *J. Phys. Chem. C* 2014, **118**, 7569–7578.), especial for the ultraviolet and blue OLEDs. Then, hydrocarbons aromatic emitters also had a intrinsically stability to resist high band exciton for stable ultraviolet and blue OLEDs, benefits of the removal of heteroatoms to avoid the weak and chemical bond in the chromophore, similar to the commercialized anthracene (*Nat Commun* 2023, **14**, 3927. *Angew. Chem.* 2022, **134**, e202207204). To the best of our knowledge about previously reported narrowband non-doped ultraviolet OLEDs (CIE_y < 0.04), the great majority showed the EQE lower than 4%. In addition, compared to the planar and fused aromatic emitters, non-planar conjugated molecules with a multi-dimensional topical structure showed an extremely weak intermolecular aggregation and interaction to obtain single-molecular emission behavior with low ratio of non-radiative transitions (*NPG Asia Materials*, 2021, **13**(1): 53.). Therefore, it is very excited to obtain a narrowband and efficient ultraviolet non-doped OLEDs based on the novel steric hydrocarbon *cis-trans*-TWG1. Therefore, it is reasonably believed that our novel steric MNHCs have a promising application in the efficient and stable UV OLEDs.

We also provided discussion in detail about the fabrication of UV OLED in this revised manuscript as follow:

“In general, our novel rigid and steric MNHCs present pure UV emission behavior in solid states. Especially, their intrinsically multi-dimensional steric interactions may be beneficial for the suppression of intermolecular π -interaction to obtain robust emission behavior, which are useful to manufacture the efficient non-doped UV organic light-emitting diodes (OLEDs). As discussion above, TWG coated films had a relatively high UV emission efficiency than those of the DWG ones, so we set *cis-trans*-TWG for optoelectronic application. Therefore, to verify their high potential application for use in electroluminescent device, non-doped OLEDs are fabricated with a configuration of ITO/ MoO₃/ mCP/Nanogrids EML/HEL/LiF/Al (120 nm) (Fig. 5e). Here, molybdenum oxide (MoO₃) serves as the hole injection layer, 9,9'-(1,3-Phenylene)bis-9H-carbazole (mCP) and 1,3,5-Tris(3-pyridyl-3-phenyl) benzene (Tmppyb) are employed as the hole and electron transport layers, respectively, and lithium fluoride (LiF) acts as a buffer layer to enhance the device's electron injection capability. *cis-trans*-TWG1 is utilized as the emitting layer (EML). Interestingly, **Device 1** based on *cis-trans*-TWG1 achieved the excellent color purity with a pure UV maximum EL emission peaked at 370 nm with narrow full width at half maximum (FWHM) as small as 34 nm, and corresponding to the ideal CIE coordinate of (0.166, 0.047), locate at the near-UV edge of the chromaticplane (Fig. 5f). These results also effectively confirmed the promising applicaiton of our novel streic MNHCs in UV non-doped OLEDs.

Meanwhile, we further optimized *cis-trans*-TWG1-based OLEDs structure trying to improve the device efficiency, observed that the ETL can slightly change the EL peak wavelength and FWHM, further slightly change the color purity, due to the optical microcavity effect (Fig. 5f)⁵⁶. The device efficiency is significantly influence by the electron affinity of the ETL and emitter materials, which may modulate the charge injection and recombination in the EML. According to the energy level as displayed in the Fig. 5e, the selection of mCP and Tmpypb as HTL and ETL nearly equal to *cis-trans*-TWG1 is beneficial for enhancing the barrier-free hole/electron injection and transportation. Besides, we tuned the film thickness of mCP, such as 45 nm (Device 1), and 55 nm (Device 2), as displayed in Fig. 5f, the EQE value of device are increasing from 2.32%, to 4.17% (Fig. 5g). Large recombination zone may be observed for a thicker HTL layer in the OLEDs to induce this enhancement of device efficiency (EQE). And corresponding the turn-on voltages (V_{on}) are estimated about 4.4 V, and 4.6 V with a maximum brightness of 189 cd/m², and 248 cd/m² (Fig. 5h), respectively. Therefore, as we discussion above, our novel *cis-trans*-TWG1 endowed its UV non-doped OLEDs with the state-of-the-art EQE up to 4.17% (Fig. 5g). The record EQE may be attributed to the synergistic effect of its improved out-coupling ratio and suppressed intermolecular interaction, exciton utilization efficiency, and suitable energy levels. We also believed that the efficiency of OLEDs based on our novel molecules can be further improved via systematically optimizing device structure and doping strategy. More interestingly, device efficiency (EQE) of typical UV OLEDs (EL peak \leq 400 nm) are summarized in Fig. 5i. Surprisingly, TAZ type molecules is few UV fluorescent emitter for the efficient non-doped OLED (EQE > 3.6%, CIE: 0.17, 0.04)⁵⁷, which can further improve to EQE > 4.14% need to introduce novel hole injection layers for the realization of the balance between hole and electron injection/transportation.⁵⁸⁻⁶⁰ Therefore, the EQE of our *cis-trans*-TWG1-based OLEDs is the record value achieved by traditional non-doped fluorescence UV OLEDs based on the hydrocarbons compounds, which is the first efficient UV OLEDs based on macrocycles (Fig. 5i).”

Fig. R3. Ultraviolet emission properties of hydrocarbon nanogrids. (a) PL spectra of TWG1 and DWG1 in film and crystal. (b) PLQY of 3h, 4a, TWG1 and DWG1 in the solution, film and crystal states. (c) Chiral HPLC analysis of *cis-trans*-TWG1 eluted by MeOH/DCM (4:1) using CHIRALPAK IJ column and circular dichroism spectra of the first peak (*cis-trans*-TWG1 (SSR)) and the second peak (*cis-trans*-TWG1 (RRS)). (d) CPL spectra and g_{PL} values of *cis-trans*-TWG1 (SSR) *cis-trans*-TWG1 (RRS). (e) Configuration of the *cis-trans*-TWG1-based UV OLED. (f) EL spectrum of device 1 and device 2 based on *cis-trans*-TWG1. Inset shows the corresponding coordinates on CIE1931 chromaticity plane. (g) External quantum efficiency-current density curve and (h) the current density-voltage-luminance (J - V - L) characteristics of device 1 and device 2 based on *cis-trans*-TWG1 UV OLED. (i) External quantum efficiency summary of the representative UV OLEDs (EL Peak \leq 400 nm). Blue represents non-doped traditional fluorescence OLED; Purple represents doped traditional fluorescence OLED; Green represents hybridized local and charge-transfer (HLCT) based OLEDs, Black represents thermally activated

delayed fluorescence (TADF) OLED.

In summary, the highlights presented in this work, spanning organic synthesis, fluorescence emission properties, CPL properties, and UV-OLED devices, may well meet the criteria for broad interest required for publication in *Nature Communications*.

About the novelty, the authors should clarify if this study is the first achiral catalyzed diastereoselective gridization.

Answer: Thank you for your suggestion on the novelty. For achiral catalyzed diastereoselective gridization, although diastereoselective (~ 9:1 d.r) synthesis of diazafluorene-based nanogridarene using achiral acids has been reported in our previous work (*Nat. Commun.* **11**, 1756 (2020)), it was based on the Friedel-Crafts reaction rather than C-H activation. Additionally, this method also faced limitations when applied to the synthesis of an all-carbon-based strained nanogrid. Furthermore, achieving high diastereoselectivity in C(sp³)-C coupling based on C(sp³)-H activation is highly challenging (*Chem. Rev.* **117**, 8908–8976 (2017); *Nat Rev Methods Primers* **1**, 43 (2021)), and in most cases, it requires the use of expensive chiral transition metal catalysts. However, in this study, we achieved excellent enantioselectivity (> 99:1 d.r) based on the π - π interactions between the phenyl group on the achiral ligand PPh₃ and the substrate's aromatic ring.

Additionally, the novelty of this work is demonstrated by the synergistic effect of conjugation disruption and steric hindrance functionalization strategies, enabling MNHCs UV fluorescence emission (displaying UV luminescence in solution, thin film, and single crystal states). In contrast, conventional MNHCs struggle to achieve UV emission, even in dilute solution, due to high strain energy, extended π -conjugation, and strong intermolecular π - π interactions. To further enhance the novelty of this work, we supplemented the chiroptical properties of *cis-trans*-TWG1 and the UV OLED devices of *cis-trans*-TWG1 and meso-DWG1. In terms of chiroptical properties, based on the novel MNHCs structure of *cis-trans*-TWG1, a high luminescence dissymmetry factor (0.01) was achieved, surpassing the luminescence dissymmetry factors of most organic chiral small molecules (at the level of 10⁻⁴ to 10⁻³). This provides a new avenue for designing and synthesizing organic small molecules with high luminescence dissymmetry factors.

More importantly, as anticipated, efficient and narrowband UV OLEDs have been successfully fabricated with the strongest emission peak at 370 nm and the highest EQE (external quantum efficiency) of 4.17%, comparable to UV TADF (thermally activated delayed fluorescence) devices. To the best of our knowledge, in previously reported narrowband non-doped ultraviolet OLEDs (peak \leq 370 nm, CIE_y \leq 0.04), the vast majority exhibited EQEs lower than 4%. **This work signifies the first application of macrocyclic structures in UV-OLEDs, and the EQE of 4.17% sets a record for the efficiency of UV OLEDs based on pure hydrocarbon emitters.**

Another comment is more details about the methods should be provided, such as the way to

calculate the transition states.

Answer: Thank you for your careful comments and suggestions.

DFT calculations. All the quantum chemical calculations were carried out using the Gaussian 09 suite of programs. Hybrid-density functional theory (DFT) calculations with the B3LYP and 6-31G(d) basis set were used to explore the geometrical structure and electronic properties of Ph, *meso*-DWG1, *rac*-DWG1, DPh, DF, TF, SWG-F, Th, DTh, TWG-Th, 2, 3, *cis-trans*-TWG1, *cis-cis*-TWG1, TWG-TS1 and TWG-TS2. We use the energy difference method to calculate the tensile energy by comparing the energy difference between the compact and relaxed configurations. Specifically, the tensile energy is equal to the energy of the compact configuration minus the energy of the relaxed configuration. Moreover, the geometries of transition states are optimized by DFT with B3LYP hybrid function, and 6-31G(d) basis sets and Los Alamos effective core potential basis set (LANL2DZ) are utilized for C, H, P atoms and Pb atom. The harmonic vibrational frequency analyses were performed on the transition states (one imaginary frequency) at the same level of theory. The circular dichroism spectrum of *cis-trans*-TWG1(SSR) and *cis-trans*-TWG1(RRS) were simulated by TD-DFT at the CAM-B3LYP/6-31G(d) level.

Fabrication of OLEDs. The ITO substrates were successively cleaned in ultrasonic bath of acetone, isopropanol, detergent and deionized water, respectively. After totally drying in a 70 °C oven, the ITO substrates were treated by O₂ plasma to improve the hole injection ability. The vacuum-deposited OLEDs were fabricated under a pressure of less than 5×10^{-4} Pa in the Fangsheng OMV-FS380 vacuum deposition system. The organic layers, LiF and Al were deposited at rates of 1-2, 0.1, and 5 Å s⁻¹, respectively.

Measurement of OLEDs. The active area of each device was 3 mm × 3 mm. The L-V-J characteristics and EL spectra of UV OLEDs were measured by a Konica Minolta CS-200 Color and Luminance Meter and an Ocean Optics USB 2000+ spectrometer, along with a Keithley 2400 Source Meter. For OLEDs, the L-V-J characteristics, EL spectra were obtained via a PhotoResearch PR670 spectroradiometer, with a Keithley 2400 Source Meter. The EQEs were estimated assuming that the devices are Lambertian emitters. All characterizations were done at room temperature under ambient conditions without any encapsulation.

Reviewer #3:

In this manuscript, Wei et al reported a stereoselective Csp²-Csp³ coupling gridization based on the C-H-activation of fluorene and synthesized dimeric and trimeric windmill-type nanogrids. The resultant nanogrids show ultraviolet emission behaviors and crystal-induced luminescence enhancement features. Combining the experimental results and calculations, the authors have clearly revealed the gridization rules, stereoselective origin, and molecular strain of resultant nanogrids.

The manuscript may be accepted given the authors address the points detailed below.

1. The products rac-DWG1, cis-trans-TWG1 and cis-cis-TWG1 have the clear chiral features and the separation seems promising in supplementary fig. 2. It is essential to report their chiroptical properties, including circular dichroism, circularly polarized luminescence properties and individual dissymmetric factors. Their unique chiroptical properties might attract widespread readership in macrocycles fields.

Answer: Thank you for your careful comments and suggestions. In accordance with your suggestions, we successfully separated cis-trans-TWG1 pure samples instead of a mixture of *cis-trans*-TWG1 diastereomers using chiral HPLC with MeOH/DCM=80/20(V/V) as the mobile phase, resulting in two distinct peaks (P1 and P2, Table R1 and Fig. R4.). Comparative analysis based on measured circular dichroism (CD) spectra and simulated data indicates that P1 corresponds to *cis-trans*-TWG1 (*SSR*), while P2 corresponds to *cis-trans*-TWG1 (*RRS*). Considering the fluorescence emission properties of *cis-trans*-TWG1, we further characterized the circularly polarized luminescence properties of the enantiomers *cis-trans*-TWG1 (*SSR*) and *cis-trans*-TWG1 (*RRS*) in toluene solution. The results reveal strong circularly polarized luminescence intensity for both *cis-trans*-TWG1 (*SSR*) and *cis-trans*-TWG1 (*RRS*). Notably, the $|g_{PL}|$ values of *cis-trans*-TWG1 exceed 0.01, surpassing the majority of organic chiral small molecules (typically at the $10^{-4} \sim 10^{-3}$ level). This opens up new avenues for the design and synthesis of organic chiral small molecules with high luminescent dissymmetry factors, particularly those exhibiting ultraviolet luminescent properties.

Table R1 | Separation method of *cis-trans*-TWG1 in HPLC.

Column	: CHIRALPAK IJ
Column size	: 5.0 cm I.D. × 25 cm L, 10 μ m
Mobile phase	: MeOH/DCM=80/20(V/V)
Flow rate	: 30 ml/min
Wave length	: UV 225 nm
Temperature	: 38 °C

Fig. R4. Chiral resolution of *cis-trans*-TWG1 and characterization of its chiral optical properties. (a) and (b) Chiral HPLC analysis of *cis-trans*-TWG1 eluted by MeOH/DCM (4:1) using CHIRALPAK IJ column. (c) Simulated circular dichroism spectra of *cis-trans*-TWG1(SSR) and *cis-trans*-TWG1(RRS). (d) Experimental circular dichroism spectra of the first peak (*cis-trans*-TWG1 (SSR)) and the second peak (*cis-trans*-TWG1 (RRS)). (e) CPL spectra and (f) g_{PL} values of *cis-trans*-TWG1 (SSR) *cis-trans*-TWG1 (RRS).

2. Although the products are characterized by NMR with assignments by 2D NMR and crystal structures, the purity of the products is important. In SI, page S29, figure S9, there are obvious impurity showing the peaks around 10-50 ppm. The authors emphasize that the nanogrids present first reported UV emission among macrocycles. It is especially important to have pure sample for optical properties measurements and avoid errors in reporting the properties of this family of nanogrids.

Answer: Thank you for your careful comments on the purity of our products. We have removed residual impurities from the samples by recrystallization using refluxed petroleum ether and dichloromethane. The purified sample's ^{13}C -NMR spectrum (Fig. R5-8.) shows no impurity peaks in the range of approximately 10-50 ppm. Moreover, the purified samples were then recharacterized for their optical properties, and the test results showed no significant differences between the further purified and previous samples. Minor discrepancies in UV/PL and PLQY data have been corrected. Additionally, both chiroptical properties and UV-OLED devices were evaluated using the purified samples. These fantastic optoelectronic properties of our novel molecules also effectively confirmed their greatly potential application in the organic photonic and electronics. We also provided a discussion in detail about their photophysical and electrical property in this revised manuscript. We sincerely hope that these changes and optimizations meet your requirements.

Fig. R5. ^{13}C -NMR spectra of *cis-trans*-TWG1.

Fig. R6. ¹³C-NMR Spectra of *cis-cis*-TWG1.

Fig. R7. ¹³C-NMR Spectra of *meso*-DWG1

Fig. R8. ¹³C-NMR Spectra of *rac*-DWG1.

3. The manuscript and SI might need to be polished. It is not easy to follow their expression. In addition, there are some mistakes in the manuscript that should be corrected. For example, in the section of investigation of gridization rules, “we found that a concentration of 40 mM resulted in a 32% yield of TWGs1 with 82:18”. In Supplementary Table 1, the result should be 35% yield with Dr ratio of 80:20. In the same section, “Substituting *t*-BuOK with KN(SiMe₃)₂ did not significantly affect the yield and selectivity of TWG1(Supplementary Table 1, entry 13), while no TWG1 was observed by using KOH (Supplementary Table 1, entry 14).” According to Supplementary Table 1, it should be entry 14 and 15, respectively.

Answer: Thank you for your kind suggestions. According to your comments, we have corrected and modified improper explanation and sentence in this revised manuscripts. I'm sorry for the confusion caused by our oversight in filling out the data in the SI. We have corrected the data for entry 8 in Supplementary Table 1 to 32% yield and a ratio of 82:18. Additionally, as we provided a separate explanation for substrate 1b (2-bromo-9-octyl-9H-fluorene) in the main text, its data is not included in Supplementary Table 1. However, we failed to correct this oversight in a timely manner. We have now removed the data, and the original entry 14 and entry 15 have been corrected to entry 13 and entry 14. We sincerely hoped that these change and optimization will meet your requirement.

4. Please replace the supplementary fig. 5-8 with high-resolution mass spectra.

Answer: Thank you for your kind suggestions. We have replaced the supplementary fig. 5-8 with high-resolution mass spectra. We sincerely hoped that these change will meet your requirement.

Other changes

1. Authors, Page 1, line 5: Because the professional and substantial contribution about the physical property in revised manuscript, we added two new authors, Jinyi Lin (as new corresponding author) and Mingjian Ni.
2. Contact address, Page 1, line 14: We added the “²Key Laboratory of Flexible Electronics (KLOFE) & Institute of Advanced Materials (IAM), Nanjing Tech University (NanjingTech), 30 South Puzhu Road, Nanjing 211816, China.”
3. Contact address, Page 1, line 19: We added the “⁵School of Flexible Electronics (SoFE) and Henan Institute of Flexible Electronics (HIFE), Henan University, 379 Mingli Road, Zhengzhou 450046, China.”
4. Abstract, Page 1, line 24: “cross-scale materials by facilitating the generation and their networks of” has been changed to “novel multifunctional organic semiconductors, such as”.
5. Abstract, Page 1, line 32: “nanogrids” has been changed to “stereo-nanogrids”.
6. Abstract, Page 1, line 34: “representing the first report of ultraviolet-emitting macrocycles” has been changed to “and circularly polarized luminescence with high luminescent dissymmetry factors ($|g_{PL}|=0.01$)”.
7. Abstract, Page 1, line 34: “diodes and lasing” has been changed to “optoelectronic”.
8. Introduction, Page 2, line 45: “Furthermore, the macrocyclization feature of dynamic covalent chemistry has led to significant advances in 2D/3D covalent organic frameworks. However, the stability requirements of these materials remain a significant challenge. Additionally, there is also a competition between linear polymerization and macrocyclization. These create numerous opportunities for developing effective macrocyclization strategies that offer diverse multiscale structures with 0/1/2/3 dimensions” has been changed to “However, due to the strain energy, extended π -conjugation and intense intermolecular π - π interactions, these MNHCs face challenges in achieving ultraviolet (UV) luminescence, limiting their exploration in the field of UV emission¹³⁻¹⁶. Additionally, the competition between linear polymerization and macrocyclization has also made the synthesis of MNHCs more challenging. These provides numerous opportunities for developing efficient macrocyclization strategies to synthesize MNHCs with specific properties”.
9. Introduction, Page 2, line 52: In the past, gridization offered one unique macrocyclization by utilizing the 2,9-position of fluorene to create the closed structures that allow for the potential linkage of 7-position sites with each other. As a result, organic nanogridarenes, a new class of C(sp³)-linked macrocycles-like compounds, have been created with great potential for extension along well-defined edges and vertices to design cross-scale molecular nanosystems.¹⁵ By designing the linkage at the 2/9-position of fluorenes, several nanogrids¹⁶⁻¹⁸, multi-grids¹⁹, and even organic nanopolymer material systems^{15,20,21} have been synthesized via Friedel-crafts gridization (FCG). Although superelectrophile-assisted FCG has achieved a stereoselective control of diazafluorene-based Drawing Hand Grids,²² achieving stereoselective synthesis of all carbon-based nanogrids via FCG remains a challenge. Furthermore, FCG exhibits limited substrate scopes owing to the electrophilic reaction feature that is mainly suitable for building blocks containing electron-rich groups. It has been difficult to directly connect the 2/9-position

of fluorene to form smaller sub-nanometer nanogrids” has been changed to “Differing from the the conventional macrocyclization methods employed in the construction of covalent organic frameworks^{8,9,17}, two-dimensional materials¹⁸, and classical macrocycles¹⁹, gridization represent a novel macrocyclization approach to building organic semiconductors. This method offers advantages such as flexibility, simplicity, and scalability, providing benefits not found in traditional macrocyclization and fusion cyclization. Until now, the achievement of gridization involves the sophisticated utilization of the 2/9 position of fluorene, thereby creating nano-grids that enable the potential linkage of 7-position sites. Significantly, through the application of the Friedel-Crafts gridization strategy, various C(sp³)-linked nanogrids²⁰⁻²², multi-grids²³, and even organic nanopolymer material systems²⁴⁻²⁶ have been successfully synthesized. However, the introduction of molecular fragments containing thiophene and carbazole, leads to the fluorescence emission of the nanogrid primarily into the visible light range. Conversely, the direct connection of all carbon-based strained sub-nanometer nanogrids at the 2,9-positions of fluorene, where the conjugation is interrupted, is anticipated to realize the UV emission. Furthermore, despite the successful stereoselective control of diazafluorene-based Drawing Hand Grids achieved by superelectrophile-assisted FCG²⁷, accomplishing stereoselective synthesis of all carbon-based strained nanogrids via FCG remains a challenge”.

10. Introduction, Page 2, line 67: “novel” has been changed to “UV-luminescent”.
11. Introduction, Page 2, line 70: We removed “achieving”.
12. Introduction, Page 2, line 70: We added the “This is particularly achievable by flexibly tuning the metal species and ligands, specifically employing chiral transition metal²⁹/ligand³⁰-controlled C-H activation reactions”.
13. Introduction, Page 2, line 73: We removed the “but more economical and attractive. Additionally, C-H activation provides a potential route to Csp²-Csp³ gridization by linking fluorene molecules with each other²⁶, which is difficult to achieve through FCG. ”
14. Introduction, Page 2, line 73: “4:1” has been changed to “4.3:1”
15. Introduction, Page 3, line 90: We added the “Additionally, *cis-trans*-TWG1 exhibits a high luminescent dissymmetry factors ($|g_{PL}| = 0.01$) in circularly polarized luminescence (CPL) compared to other chiral organic small-molecules. Notably, the non-doped ultraviolet organic light-emitting diode (OLED) based on the traditional fluorescence emission mechanism of *cis-trans*-TWG1 not only demonstrates a high external quantum efficiency (EQE) of 4.17% but also has an emission peak wavelength in the UV region at 386 nm with a CIE of (0.17, 0.04), representing the first report of UV-OLED macrocycles. ”
16. Introduction, Page 3, line 90: We removed the “These nanogrids exhibit the potential applications of UV organic light-emitting diodes (OLED) and lasing.”
17. Investigation of gridization rules, Page 3, line 94: “1a” has been changed to “**1a**”.
18. Investigation of gridization rules, Page 3, line 119: “4f” has been changed to “**4f**”.
19. Stereoselective origin, Page 4, line 134: “product” has been changed to “intermediate”.
20. Stereoselective origin, Page 4, line 152: We added the “of larger than trimeric nanogrids”.
21. Ultraviolet emission properties of DWGs and TWGs, Page 6, line 244: “**DWGs and TWGs**. Encouraged by the successful application of bi(9,9-diarylfuorene)s in UV-OLED,⁴⁰⁻⁴²” has been

changed to “**hydrocarbon nanogrids**. After confirming the molecular structure”

22. Ultraviolet emission properties of DWGs and TWGs, Page 7, line 261: “both DWG1 and TWG1 showed gridization-induced luminescence enhancement feature, with higher photoluminescence quantum yield (PLQY) in the nanogrids (20% for *meso*-DWG1, 15% for *rac*-DWG1, 17% for *cis-trans*-TWG1, 25% for *cis-cis*-TWG1) than in the corresponding linear molecules (6% for **3h**, 14% for **4a**) (Fig. 5f). This may be because the robust backbone and intramolecular interactions of strained nanogrids inhibit the non-radiative relaxation⁴⁴.” has been changed to “both DWG1 and TWG1 showed gridization-induced luminescence enhancement feature, with higher photoluminescence quantum yield (PLQY) in the nanogrids (21% for *meso*-DWG1, 17% for *rac*-DWG1, 25% for *cis-trans*-TWG1, 17% for *cis-cis*-TWG1) than in the corresponding linear molecules (8% for **3h**, 9% for **4a**) (Fig. 5b). This may be because the robust and steric multi-dimensional topological structure to suppress intermolecular interactions of strained nanogrids and further inhibit the non-radiative relaxation⁴⁶”.
23. Conclusion, Page 7, line 299: “the first reported UV emission among macrocycles” has been changed to “CPL property ($|g_{PL}| = 0.01$)”.
24. Conclusion, Page 7, line 299: We added the “More interestingly, efficient ultraviolet OLEDs with a narrowband emission at 386 nm are also fabricated with a highest EQE of 4.17%, representing the first report of UV-OLED macrocycles”.
25. Conclusion, Page 7, line 299: We removed the “and lasing”.
26. Supplementary Section 1, Characteristics of C-H gridization, page 17: We removed the “**Supplementary Fig. 3 | The CPLC spectrum of TWGs, the peak area ratio of *cis-cis*-TWG1 to *cis-trans*-TWG1 is 7.9% : 92.1%.** The mother liquor of recrystallization was processed by cycle preparation liquid chromatography (CPLC) using dichloromethane as the mobile phase. In the CPLC, *cis-cis*-TWG1 (shaded in green) and *cis-trans*-TWG1 (shaded in blue) were successfully separated and prepared when the retention time reaches 700 min.”

REVIEWER COMMENTS

Reviewer #1 (Remarks to the Author):

The reviewer acknowledges the sincere efforts of these authors to respond to my comments. As a result of their efforts, the computational process of the cyclization mechanism seems to be organized and some interesting insights into the physical properties of the cyclization products have been found. However, as will be discussed later, there are some questions about the chiroptical properties. On the other hand, the high performance for OLEDs is an interesting result. For each of the newly added cyclization product properties, a little more research background needs to be given in the introduction to each topic.

As mentioned above, there are several questions regarding the chiroptical properties. The authors report a g -value as high as 0.01 in CPL for the compound *cis-trans*-TWG1. However, a careful comparison of the spectra in Figure 5d and Supplementary Fig. 29 shows that there is almost no emission at wavelengths with a g -value of 0.01. Near the maximum peak of emission, the value is about 0.005, which is probably in the average range of values for chiral macrocycles. Also, the solvent should be given in the absorption and emission spectra, as well as in the CD and CPL spectra. Emission and CPL spectra should be shown side by side. The same is true for the absorption and CD spectra. For the CD spectra in Figure 5c, the SSR and RRS spectra are similar, but the peaks have different shapes and do not form a mirror image. This is very strange. The simulation of the CD spectrum in Supplementary Fig. 46 is also not a mirror image, which is confusing to the reviewer.

Overall, this reviewer acknowledges the authors' efforts and commend them for finding interesting physical properties. However, the discussion of chiroptical properties is still immature and needs to be improved. It is difficult for this reviewer to state at this time whether this study is worthy of publication in this journal.

Reviewer #2 (Remarks to the Author):

The authors mentioned that they achieved a record EQE value of 4.17%. Please specify how much improvement is there compared to the highest previous value. If it is only small improvement, it may still not be able to justify the publication in NC.

Reviewer #3 (Remarks to the Author):

In the revised manuscript, the authors have purified the diastereomers and the interesting optical properties and chiroptical properties have been provided, including enhanced photoluminescence quantum yields and excellent luminescent dissymmetry factors (0.01) surpassing the 10⁻³-10⁻⁴ range. In addition, the resultant products have been used in OLED, presenting high external quantum efficiency,

which hold great potential in the organic photonic and electronics. The reviewer's concerns have been clarified; therefore, the reviewer would recommend the publication of this work in nature communication.

Respond to Reviewers

We sincerely thank the reviewers for carefully reviewing our work and the constructive and insightful comments, which have helped us to improve the paper further. Our detailed responses to the reviewer's comments are provided in the following. Changes in the revised manuscript as a response to the reviewers' comments are highlighted in *red color* and clarifications regarding the reviewer's comments are provided in *blue color*.

Respond to Reviewers:

Reviewer #1:

The reviewer acknowledges the sincere efforts of these authors to respond to my comments. As a result of their efforts, the computational process of the cyclization mechanism seems to be organized and some interesting insights into the physical properties of the cyclization products have been found. However, as will be discussed later, there are some questions about the chiroptical properties. On the other hand, the high performance for OLEDs is an interesting result. For each of the newly added cyclization product properties, a little more research background needs to be given in the introduction to each topic.

Answer: Thank you for your careful comments and suggestions.

According to your suggestions, we have incorporated research background of chiroptical properties and UV OLED into the introduction section. We sincerely hoped that these change and optimization will meet your requirement. We also hoped that our work can be accepted and published in Nature Communications.

As mentioned above, there are several questions regarding the chiroptical properties. The authors report a g-value as high as 0.01 in CPL for the compound cis-trans-TWG1. However, a careful comparison of the spectra in Figure 5d and Supplementary Fig. 29 shows that there is almost no emission at wavelengths with a g-value of 0.01. Near the maximum peak of emission, the value is about 0.005, which is probably in the average range of values for chiral macrocycles. Also, the solvent should be given in the absorption and emission spectra, as well as in the CD and CPL spectra. Emission and CPL spectra should be shown side by side. The same is true for the absorption and CD spectra.

Answer: Thank you for your careful comments and suggestions.

We sincerely apologize for any confusion stemming from our oversight in clearly specifying the testing conditions for CPL and fluorescence spectra. The absorption and fluorescence emission spectra were recorded using tetrahydrofuran as the solvent at a concentration of

10^{-5} M. Conversely, the CD and CPL spectra were obtained using toluene as the solvent at a concentration of 5×10^{-6} M. These conditions have been explicitly stated in the revised manuscript. Due to solvent discrepancies, there are variations in the peak positions of the fluorescence emission spectra. When aligning with the CPL spectra conditions, the fluorescence emission peaks of *cis-trans*-TWG1 (SSR) and *cis-trans*-TWG1 (RRS) were identified at 348 nm (Fig. R1). When there is a disparity between the peak positions of the emission spectrum and the CPL spectrum, the g-value was calculated based on the CPL spectra's peak position (343 nm) and determined to be 0.01 (*Nat Commun*, **2023**, 14, 8022; *Angew. Chem. Int. Ed.* **2020**, 59, 4756-4762; *Nat Commun*, **2017**, 8, 15727). Furthermore, in response to your request, the absorption and CD spectra have been redrawn (Fig. R2). We sincerely hoped that these change and optimization will meet your requirement.

Fig. R1. The fluorescence emission and CPL spectra of *cis-trans*-TWG1 (SSR) and *cis-trans*-TWG1 (RRS) in toluene solution ($C = 5 \times 10^{-6}$ M).

Fig. R2. The absorption and CD spectra of *cis-trans*-TWG1 (SSR) and *cis-trans*-TWG1 (RRS) in toluene solution ($C = 5 \times 10^{-6}$ M).

For the CD spectra in Figure 5c, the SSR and RRS spectra are similar, but the peaks have different shapes and do not form a mirror image. This is very strange. The simulation of the CD spectrum in Supplementary Fig. 46 is also not a mirror image, which is confusing to the reviewer.

Answer: Thank you for your careful comments and suggestions.

We are agreed with your comment about the mirror images of the SSR and RRS under the ideal condition and theoretical calculation. However, as far as we know, it's not unusual for enantiomeric pairs' CD spectra to exhibit slight differences in peak shape and discrepancies in mirror image presentation due to variations in peak heights (*Angew. Chem. Int. Ed.* **2020**, *59*, 11267-11272; *Adv. Mater.* **2022**, *34*, 2105080; *Adv. Mater.* **2021**, *33*, 2100652; *J. Am. Chem. Soc.* **2021**, *143*, 18527–18535). To simulate CD spectra, we employed DFT calculations with a higher precision basis set (CAM-B3LYP/TZVP), yielding simulated CD spectra that mirror each other (Fig. R3). We sincerely hoped that these change and optimization will meet your requirement.

Fig. R3. Simulated circular dichroism spectra of *cis-trans-TWG1(SSR)* and *cis-trans-TWG1(RRS)* based on the basis set CAM-B3LYP/TZVP.

Overall, this reviewer acknowledges the authors' efforts and commend them for finding interesting physical properties. However, the discussion of chiroptical properties is still immature and needs to be improved. It is difficult for this reviewer to state at this time whether this study is worthy of publication in this journal.

Answer: Thank you for your careful comments.

In response to your guidance, we've revised sections identified as problematic and bolstered the discourse on chiroptical properties. Specifically, we've added comprehensive annotations detailing the testing conditions for chiroptical properties and recalibrated the CD spectra using a more precise basis set. We sincerely hoped that these change and

optimization will meet your requirement.

Finally, we are sincerely hoped that our respond can be met your high quality requirement and you can agree to accept our manuscript to publish in *Nature Communications*.

Reviewer #2:

The authors mentioned that they achieved a record EQE value of 4.17%. Please specify how much improvement is there compared to the highest previous value. If it is only small improvement, it may still not be able to justify the publication in NC.

Answer: Thank you for your careful comments and suggestions.

As your comment, our device present a substantially improvement in EQE, compared to those of device based on hydrocarbons. Meanwhile, we also need to note that our device performance can be further improved after systemic optimization. Firstly, in comparison to the reported highest EQE value of 3.6% for UV OLEDs based on hydrocarbons (Table R1), the peak EQE value for UV OLEDs employing *cis-trans*-TWG1 in this study reaches 4.17%, signifying a notable and substantial performance enhancement of 15.83%. Secondary, in general, it's typical for emitters with shorter emission wavelengths (wide band-gap) to exhibit decreased EQE value due to the synergistic effect of large exciton energy, low charge density and mobility. However, despite a 6 nm blue shift in the emission peak position compared to the devices of B2, UV OLEDs utilizing *cis-trans*-TWG1 still demonstrate a 15.83% improvement in EQE. Finally, the theoretically maximum EQE value for OLEDs relying on conventional fluorescent emission mechanisms is approximately 5% (*Adv. Funct. Mater.* **2023**, 2312622; *Adv. Optical Mater.* **2018**, 6, 1800512; *Adv. Mater.* **2014**, 26, 7931-7958), suggesting that this study elevates the performance of UV OLED devices based on hydrocarbons from around 72% (B2) of the theoretical maximum EQE to about 83.4% (our novel molecules, *cis-trans*-TWG1).

In summary, the comparative results underscore a significant performance enhancement in the maximum EQE value of 4.17% in this study compared to the highest reported value of 3.6%, potentially rendering this research suitable for publication in *Nature Communications*.

Table R1 Performance summary of UV OLEDs based on hydrocarbons (EL Peak \leq 400 nm)

UV emitter	λ_{EL} (nm)	EQE _{max} (%)	FWHM (nm)	Ref.
B2	392	3.6	-	Adv. Mater. 2005 , 34, 992-996
TB2	396	2.7	-	Adv. Mater. 2005 , 34, 992-996
1SBFN	385	2.9	-	Appl. Phys. Lett. 2001 , 79, 2282–2284
2SBFN	382	2.2	-	Appl. Phys. Lett. 2001 , 79, 2282–2284
BSBFB	388	1.6	-	Appl. Phys. Lett. 2001 , 79, 2282–2284
TSBFB	395	2.1	-	Appl. Phys. Lett. 2001 , 79, 2282–2284
DPPP	396	2.2	50	Org. Lett. 2005 , 7, 5131-5134
SSS	393	2.6	-	Opt. Mater. 2020 , 108, 110159
cis-trans -TWG1 (Device 2)	386	4.17	49	This work

Finally, we are sincerely hoped that our respond can be met your high quality requirement and you can agree to accept our manuscript to publish in *Nature Communications*.

Reviewer #3:

In the revised manuscript, the authors have purified the diastereomers and the interesting optical properties and chiroptical properties have been provided, including enhanced photoluminescence quantum yields and excellent luminescent dissymmetry factors (0.01) surpassing the 10^{-3} - 10^{-4} range. In addition, the resultant products have been used in OLED, presenting high external quantum efficiency, which hold great potential in the organic photonic and electronics. The reviewer's concerns have been clarified; therefore, the reviewer would recommend the publication of this work in nature communication.

Answer: Thank you for your valuable comments and suggestions, which have helped further improve the quality of the paper to meet the standards for publication in Nature Communications. We also appreciate your recognition of our work.

REVIEWER COMMENTS

Reviewer #1 (Remarks to the Author):

Comments are noted in the attached PDF file.

Reviewer #2 (Remarks to the Author):

The revised manuscript is recommended to be published in NC.

The presentation of spectra, including CD and CPL spectra, still seems immature.

Fig. R1 appears to correspond to Fig. 5d, but the figure is not corrected in the main text PDF file. Additionally, as shown in the figure below, the CPL and PL figures are truncated in the middle of the peaks. Please show the entire wavelength region so that all peaks are visible. The excitation wavelength should be included.

The CPL and PL figures are truncated in the middle of the peaks.

Fig. R1. The fluorescence emission and CPL spectra of *cis-trans*-TWG1 (SSR) and *cis-trans*-TWG1 (RRS) in toluene solution ($C = 5 \times 10^{-6}$ M).

The excitation wavelength should be included.

Regarding Fig. 5C (= Fig. R2), this reviewer made the following comments “For the CD spectra in Figure 5c, the SSR and RRS spectra are similar, but the peaks have different shapes and do not form a mirror image. This is very strange.”. The authors responded as follows “However, as far as we know, it's not unusual for enantiomeric pairs' CD spectra to exhibit slight differences in peak shape and discrepancies in mirror image presentation due to variations in peak heights (Angew. Chem. Int. Ed. 2020, 59, 11267-11272; Adv. Mater. 2022, 34, 2105080; Adv. Mater. 2021, 33, 2100652; J. Am. Chem. Soc. 2021, 143, 18527–18535).” I have a feeling of discomfort regarding the points with different shapes of peaks as indicated in the figure below. While the authors try to dismiss the issue by saying that CD spectra not being mirror images is not unusual, considering that it's not just the intensity of the peaks but the different shapes, I believe it's prudent to exercise caution in this matter. I understand the desire to be published quickly, but I suggest taking the time to confirm the purity of the samples and remeasuring the CD spectrum.

Effect of impurity? Or not an enantiomeric pair?

Fig. R2. The absorption and CD spectra of *cis-trans-TWG1 (SSR)* and *cis-trans-TWG1 (RRS)* in toluene solution ($C = 5 \times 10^{-6}$ M).

Respond to Reviewers

We sincerely thank the reviewers for carefully reviewing our work and the constructive and insightful comments, which have helped us to improve the paper further. Our detailed responses to the reviewer's comments are provided in the following. Changes in the revised manuscript as a response to the reviewers' comments are highlighted in *red color* and clarifications regarding the reviewer's comments are provided in *blue color*.

Respond to Reviewers:

Reviewer #1:

The presentation of spectra, including CD and CPL spectra, still seems immature. Fig. R1 appears to correspond to Fig. 5d, but the figure is not corrected in the main text PDF file. Additionally, as shown in the figure below, the CPL and PL figures are truncated in the middle of the peaks. Please show the entire wavelength region so that all peaks are visible. The excitation wavelength should be included.

Answer: Thank you for your careful comments and suggestions.

Taking your advice into consideration, we commenced with an additional purification step for the sample. Subsequently, we proceeded to reevaluate its chiroptical properties using the refined sample. The g_{PL} , CPL, and fluorescence emission spectra of *cis-trans*-TWG1 (SSR) and *cis-trans*-TWG1 (RRS) were then acquired, as depicted in Fig. R1, with toluene serving as the solvent ($C = 5 \times 10^{-6}$ M) under an excitation wavelength of 290 nm. The entire wavelength region is clearly visible. Specifically, the fluorescence emission peaks for *cis-trans*-TWG1 (SSR) and *cis-trans*-TWG1 (RRS) were detected at 347 nm, whereas their CPL spectra peaked at 340 nm. The g -value was calculated based on the CPL spectra's peak position (340 nm) and determined to be 0.012. Moreover, the Fig. 5d has been corrected in the revised manuscript. We sincerely hoped that these change and optimization will meet your requirement.

Fig. R1. The g_{PL} , CPL and fluorescence emission spectra ($\lambda_{ex} = 290$ nm) of *cis-trans*-TWG1 (SSR) and *cis-trans*-TWG1 (RRS) in toluene solution ($C = 5 \times 10^{-6}$ M).

Regarding Fig. 5C (= Fig. R2), this reviewer made the following comments “For the CD spectra in Figure 5c, the SSR and RRS spectra are similar, but the peaks have different shapes and do not form a mirror image. This is very strange.”. The authors responded as follows “However, as far as we know, it's not unusual for enantiomeric pairs' CD spectra to exhibit slight differences in peak shape and discrepancies in mirror image presentation due to variations in peak heights (Angew. Chem. Int. Ed. 2020, 59, 11267-11272; Adv. Mater. 2022, 34, 2105080; Adv. Mater. 2021, 33, 2100652; J. Am. Chem. Soc. 2021, 143, 18527–18535)”. I have a feeling of discomfort regarding the points with different shapes of peaks as indicated in the figure below. While the authors try to dismiss the issue by saying that CD spectra not being mirror images is not unusual, considering that it's not just the intensity of the peaks but the different shapes, I believe it's prudent to exercise caution in

this matter. I understand the desire to be published quickly, but I suggest taking the time to confirm the purity of the samples and remeasuring the CD spectrum.

Answer: Thank you for your careful comments and suggestions.

Taking your advice into consideration, we commenced with an additional purification step for the sample. Subsequently, we proceeded to reevaluate its CD spectra using the refined sample. The characterization results reveal that the CD spectra of *cis-trans*-TWG1 (SSR) and *cis-trans*-TWG1 (RRS) display mirror images (Fig. R2), despite the existence of subtle differences. Moreover, the Fig. 5c has been corrected in the revised manuscript. We sincerely hoped that these change and optimization will meet your requirement.

Fig. R2. The absorption and CD spectra of *cis-trans*-TWG1 (SSR) and *cis-trans*-TWG1 (RRS) in toluene solution ($C = 5 \times 10^{-6}$ M).

Finally, we are sincerely hoped that our respond can be met your high quality requirement and you can agree to accept our manuscript to publish in *Nature Communications*.

Reviewer #2:

The revised manuscript is recommended to be published in NC.

Answer: Thank you for your previously valuable comments and suggestions, which have helped further improve the quality of the paper to meet the standards for publication in Nature Communications.

REVIEWERS' COMMENTS

Reviewer #1 (Remarks to the Author):

The revised manuscript has shown improvement in the presentation of the CD and CPL spectra. Now, this reviewer recommends its publication in Nature Communications.